# OSTQuant: Refining Large Language Model Quantization with Orthogonal and Scaling Transformations for Better Distribution Fitting

**Xing Hu**[1*], **Yuan Cheng**[1,2*†], **Dawei Yang**[1* ✉], **Zukang Xu**[1], **Sifan Zhou**[1,3†], **Jiangyong Yu**[1]
**Chen Xu**[1], **Zhixuan Chen**[1], **Zhe Jiang**[3], **Zhihang Yuan**[1 ✉]

[1]Houmo AI    [2]Nanjing University    [3]Southeast University
xing.hu@houmo.ai, yuancheng@smail.nju.edu.cn
dawei.yang@houmo.ai, zhihang.yuan@houmo.ai

## Abstract

Post-training quantization (PTQ) has emerged as a widely adopted technique for compressing and accelerating Large Language Models (LLMs). The major challenge in LLM quantization is that uneven and heavy-tailed data distributions can expand the quantization range, thereby reducing bit precision for most values. Recent methods attempt to eliminate outliers and balance inter-channel differences by employing linear transformations; however, they remain heuristic and are often overlook optimizing the data distribution across the entire quantization space. In this paper, we introduce Quantization Space Utilization Rate (QSUR), a novel metric that effectively assesses the quantizability of transformed data by measuring the space utilization of the data in the quantization space. We complement QSUR with mathematical derivations that examine the effects and limitations of various transformations, guiding our development of Orthogonal and Scaling Transformation-based Quantization (OSTQuant). OSTQuant employs a learnable equivalent transformation, consisting of an orthogonal transformation and a scaling transformation, to optimize the distributions of weights and activations across the entire quantization space. Futhermore, we propose the KL-Top loss function, designed to mitigate noise during optimization while retaining richer semantic information within the limited calibration data imposed by PTQ. OSTQuant outperforms existing work on various LLMs and benchmarks. In the W4-only setting, it retains 99.5% of the floating-point accuracy. In the more challenging W4A4KV4 configuration, OSTQuant reduces the performance gap by 32% on the LLaMA-3-8B model compared to state-of-the-art methods. https://github.com/BrotherHappy/OSTQuant.

## 1 Introduction

Large language models (LLMs) (Dettmers et al., 2022; Touvron et al., 2023a;b) have demonstrated exceptional performance across a variety of tasks, increasingly integrating into daily life and playing critical roles in various areas (Achiam et al., 2023; Chen et al., 2024). Nevertheless, the substantial memory and computational demands pose significant deployment challenges, limiting their practical applicability not only on edge devices with constrained resources but also on cloud servers equipped with powerful GPU devices.

Post-training quantization (PTQ) has emerged as a widely adopted technique for compressing and accelerating LLMs. During quantization, uneven and heteroscedastic data, as shown in 1(a), pose significant challenges, as they expand the quantization range and reduce available bit precision for the majority of values. Recently researches (Xiao et al., 2022; Shao et al., 2023; Ma et al., 2024; Ashkboos et al., 2024) have employed linear transformations to tackle these challenges, showing

---

✉Corresponding author
*Equal contribution
†This work was conducted during his internship at Houmo

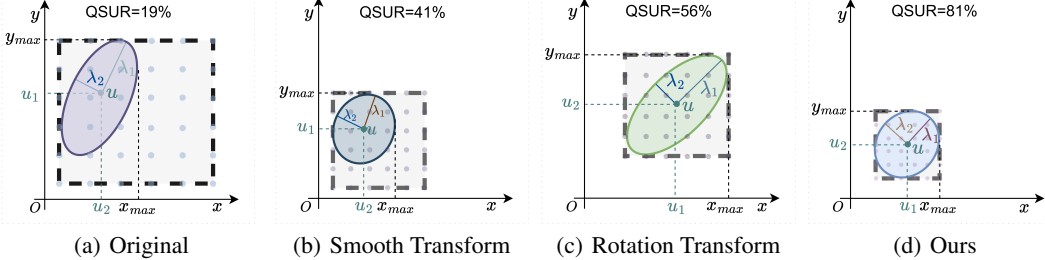

|  (a) Original | (b) Smooth Transform | (c) Rotation Transform | (d) Ours |

Figure 1: Transformation of a batch of two-dimensional data $X \sim \mathcal{N}(\boldsymbol{\mu}, \boldsymbol{\Sigma})$ using different methods. Eigenvalues $\lambda_1$ and $\lambda_2$ represent the spread of the distribution along principal axes after eigenvalue decomposition of $\Sigma$. (a) shows the original distribution, while (b), (c), and (d) illustrate the effects of the Smooth-based, Rotate-base, and ours OST-based methods, respectively, on QSUR. The ellipse represents the space occupied by the data, and the square indicates the quantization space required to quantize this distribution, determined by the maximum and minimum values of it. The gray dots within the square denote the specific quantization points within the quantization space. The higher the number of quantization points within the ellipse, the greater the quantization space utilization rate of the distribution.

effectiveness in improving data distribution in specific regions of the quantization space. For instance, smooth-based transformations like SmoothQuant (Xiao et al., 2022) alleviate activation quantization difficulty by redistributing it to weights, reducing inter-channel variance. Similarly, rotation-based methods such as Quarot use rotation techniques to suppress outliers. However, these approaches remain heuristic and do not optimize the distribution across the entire quantization space.

In this paper, we introduce the concept of Quantization Space Utilization Rate (QSUR) as a more effective metric for evaluating the quantizability of transformed data. We define QSUR as the ratio of the volume occupied by a set of data $X$ to the volume of the quantization space corresponding to $X$. Given that weights and activations typically exist in multiple dimensions, this quantization space is modeled as a hypercube where the edge lengths are determined by the maximum quantization range across all dimensions of the data. Experiments demonstrate a positive correlation between QSUR and quantization accuracy (as shown in Fig 3), suggesting that higher QSUR values contribute to improved quantization precision. In addition, we complement QSUR with mathematical derivations (refer to Sec 3 for more details), which examine the effects and limitations of linear transformations and establishes a theoretical foundation for developing more effective transformation. As shown in Fig 1, as QSUR increases, the data distribution becomes flatter, allowing more quantization levels to be utilized. Smooth-based and rotation-based transformations improve QSUR from different perspectives: smooth-based transformations excel at reducing variations among eigenvalues, whereas rotation-based transformations adeptly balance feature orientations and harmonize mean values across dimensions. However, these approaches encounter a critical limitation: their effectiveness is restricted to specific regions within the quantization space, leading to low utilization rates.

To this end, we propose Orthogonal and Scaling Transformation-based Quantization (OSTQuant). OSTQuant assigns a learnable equivalent transformation pair—comprising an orthogonal transformation and a scaling transformation—to each fully connected (FC) layer in LLMs. During optimization, these transformation pairs aim to optimize the distributions of weights and activations across the entire quantization space. The transformation pairs and their inverses are then fused into their respective FC layers, preserving the original computational graph. Besides, unlike block-wise reconstruction methods (Shao et al., 2023; Ma et al., 2024), our equivalent transformation pairs do not alter network's final output, ensuring generalization and avoiding overfitting to the calibration set.

Another challenge is that, unlike the original floating-point models trained on large datasets, PTQ is typically conducted on a small calibration set (e.g., 1000 or fewer). In this context, persisting with the original cross-entropy loss may result in a decline in model performance. To mitigate this problem, we introduce KL-Top loss function, which leverages the top $k$ highest-probability logits from the full-precision model, rather than concentrating solely on the logit probability corresponding to the correct label. Our KL-Top loss improves the capture of more nuanced semantic information while mitigating noise from the long-tail distribution in the full KL divergence. As for optimizer, we

employ RiemannAdam (James, 1976), a proficient technique tailored for learning Stiefel Manifolds using first-order information.

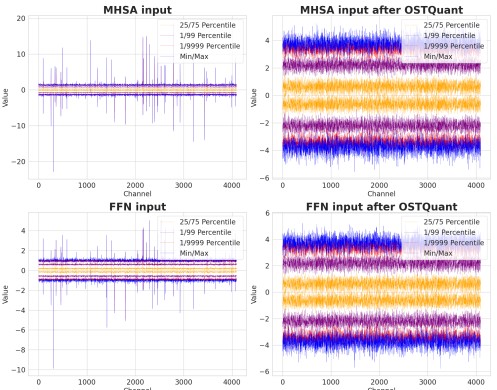
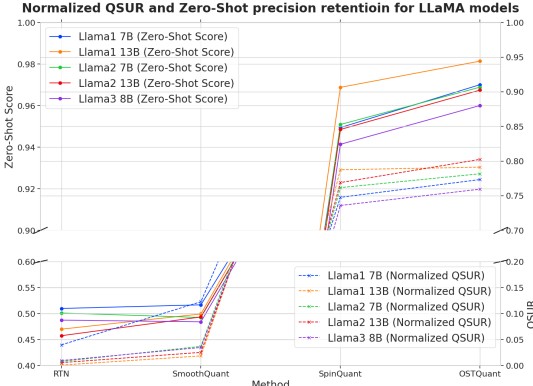

Figure 2: Activation distribution in the LLaMA-3-8B before and after applying OSTQuant shows significant differences. Prior to transformation, the distribution across different channels exhibits substantial variation and contains numerous outliers. After OSTQuant, the distributions become more uniform across channels.

Figure 3: Zero-Shot[9] precision retention (under W4A4 quantization) and normalized QSUR are evaluated for LLaMA variants across different quantization methods. The normalized QSUR is derived as the $d$-th root of QSUR, where $d$ denotes the number of channels. QSUR exhibits a positive correlation with accuracy.

OSTQuant is an effective and efficient PTQ method for LLMs, outperforming existing approaches across various models and benchmarks. In the W4-only setting, it retains above 99.5% of the floating-point accuracy. In the more challenging W4A4KV4 configuration, OSTQuant reduces the performance gap by 32% on LLaMA-3-8B compared to state-of-the-art (SOTA) methods. In terms of speed, it can complete the quantization of LLaMA-3-8B on an A800 GPU in just 20 minutes. Our contributions are summarized as follows:

- We introduce the concept of Quantization Space Utilization Rate (QSUR) as an effective metric to evaluate quantizability. We also complement QSUR with mathematical derivations that examine the effects and limitations of various transformations, guiding the next steps in optimization and method design.

- We propose OSTQuant, a fast and effective PTQ method. It improves QSUR and quantization performance by globally optimizing multiple equivalent transformation pairs in LLMs, each consisting of a scaling transformation and an orthogonal transformation. To address the challenge of limited datasets imposed by PTQ during optimization, we introduce the KL-Top loss, which captures richer semantic information from the model while mitigating the impact of label noise.

- Building on these advancements, our method demonstrates robust performance in both weight-only, weight-activation and weight-activation-kvcache quantization modes. In the W4A16 configuration, it retains over 99.5% of the full precision accuracy, while in the more aggressive W4A4KV4 setting, it maintains at least 96% of the model's original performance.

## 2 RELATED WORK

**Post Training Quantization(PTQ) for LLMs.**   Post-training quantization (PTQ) has become a mainstream technique for LLMs due to its efficiency. Existing PTQ methods can be broadly divided into weight-only and weight-activation quantization. To reduce memory usage, some approaches focus on weight-only quantization. GPTQ (Frantar et al., 2022) uses Hessian-based error compensation to achieve high compression rates by minimizing quantization errors. AWQ (Lin et al., 2023) and OWQ (Lee et al., 2023) improve performance by addressing the impact of activation outliers on weight quantization. QuIP (Chee et al., 2023) and QuIP# (Tseng et al., 2024) use random Hadamard matrices for incoherent processing and apply vector quantization to weights, achieving better performance with reduced precision quantization. Unlike weight-only methods, weight-activation quantization aims to speed up LLM inference by quantizing both weights and activations, including the key-value (KV) cache. The main challenge in activation quantization is that outliers dominate the range, leaving

few significant bits for most values, leading to substantial errors. ZeroQuant (Yao et al., 2022) proposes a fine-grained hardware-friendly quantization scheme for both weights and activations. SmoothQuant (Xiao et al., 2022) shifts quantization difficulty from activations to weights through mathematical transformation. OmniQuant (Shao et al., 2023) further enhances performance by training quantization parameters and transformation coefficients. Moreover, I-LLM (Hu et al., 2024) achieves integer-only quantization and inference through fully-smooth block reconstruction and fully integer operators. Recently, QuaRot (Ashkboos et al., 2024) uses random rotation matrices to enable 4-bit quantization of weights and activations, while SpinQuant (Liu et al., 2024) learns these matrices to refine 4-bit quantization.

**Riemannian Optimization.** The optimization of rotation matrices necessitates adherence to orthonormality constraints, which corresponds to performing Riemannian optimization on the Stiefel manifold (James, 1976), encompassing all orthogonal matrices. Cayley SGD (Li et al., 2020) relies on iterative approximations of the Cayley Transform, achieved solely by matrix multiplication, enabling effective optimization of rotation matrices for arbitrary loss functions. RAOM (Bécigneul & Ganea, 2018) extends optimization methods such as ADAM (Kingma, 2014), ADAGRAD, and AMSGRAD into the realm of Riemannian optimization. Meanwhile, Geoopt (Kochurov et al., 2020) supports fundamental Riemannian stochastic gradient descent (SGD) and adaptive optimization algorithms, facilitating seamless integration into models for comprehensive optimization.

## 3 QUANTIZATION SPACE UTILIZATION RATE

Although significant progress has been made PTQ using linear transformations to mitigate quantization loss (Xiao et al., 2022; Ma et al., 2024; Ashkboos et al., 2024), these methods are primarily heuristic and result-driven, lacking a quantitative metric to assess quantization difficulty or the effectiveness of different transformations. To address this gap, we introduce a novel metric, the Quantization Space Utilization Rate (QSUR), which quantifies how effectively weight or activation distributions utilize the available quantization space. QSUR provides critical insights into the strengths and limitations of existing methods and lays the groundwork for developing more efficient approaches, including our OSTQuant method described in Sec 4.1.

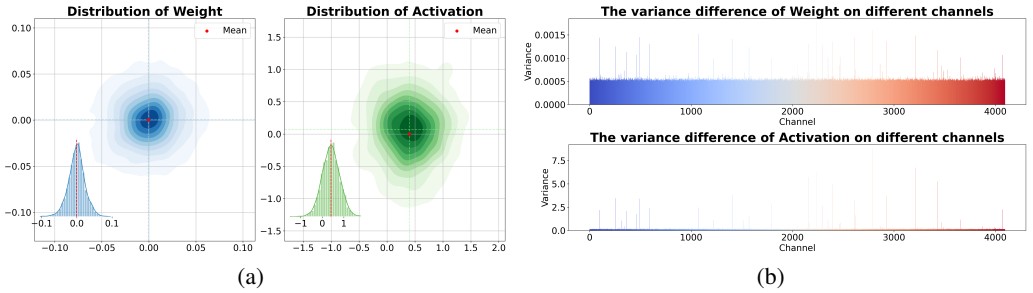

Figure 4: The distribution of activation and weight in LLaMA-2 7B. (a) The weight and activation distributions exhibit a Gaussian pattern. The red dots indicate the mean value of the distributions. Both the weights and activations are projected onto a two-dimensional space. (b) The inter-channel variance disparities between weights and activations. In comparison with weights, the inter-channel disparities of activations are more pronounced.

**Quantization Notations.** In this section, we define the key notations used in quantization. Matrices are denoted by bold uppercase letters (e.g., $\boldsymbol{X}$), while vectors are denoted by bold lowercase letters (e.g., $\boldsymbol{x}$). The operator $\mathcal{Q}$ refers to the quantization function. For a comprehensive list of mathematical symbols and definitions, please refer to Appendix A.1, where additional details on quantization and dequantization are also provided.

**Lemma 1.** *By the central limit theorem, the distribution after Hadamard transformation follows an approximately ball-shaped Gaussian distribution, as demonstrated in QuIP# (Tseng et al., 2024).*

**Definition 1.** Given a set of $d$-dimensional data $\boldsymbol{X} \in \mathbb{R}^{n \times d}$, let $V_{\boldsymbol{X}}$ denote the hypervolume occupied by $\boldsymbol{X}$, and $V_{S_{\boldsymbol{x}}}$ denote the hypervolume of the quantization space $S$ corresponding to $X$. The quantization space $S_{\boldsymbol{X}}$ is a hypercube whose edge lengths are defined by the maximum quantization range across all dimensions of $\boldsymbol{X}$. **The Quantization Space Utilization Rate** of $\boldsymbol{X}$ is then defined as:

$$\text{QSUR}_{\boldsymbol{X}} = \frac{V_{\boldsymbol{X}}}{V_{S_{\boldsymbol{x}}}} \tag{1}$$

Given $X \sim \mathcal{N}(\boldsymbol{\mu}, \boldsymbol{\Sigma})$. $V_{\boldsymbol{X}}$ is calculated based on the ellipsoid formed by the covariance matrix $\boldsymbol{\Sigma}$ and mean vector $\boldsymbol{\mu}$. The covariance matrix can be diagonalized via eigenvalue decomposition: $\boldsymbol{\Sigma} = \boldsymbol{Q}\boldsymbol{\Lambda}\boldsymbol{Q}^{\top}$, where $\boldsymbol{Q}$ is a unit orthogonal matrix of eigenvectors, and $\boldsymbol{\Lambda} = \mathrm{diag}(\lambda_1, \lambda_2, \ldots, \lambda_d)$ contains the eigenvalues in descending order. The hypervolume of this ellipsoid at confidence level $\alpha$ (e.g., $\alpha = 0.99$) is given by:

$$V_{\boldsymbol{X}} = \frac{\pi^{d/2}}{\Gamma(d/2+1)} \times \left(\chi_d^2(\alpha)\right)^{d/2} \times \sqrt{\det(\boldsymbol{\Sigma})} \tag{2}$$

where $\Gamma$ is the Gamma function and $\chi_d^2(\alpha)$ is the chi-squared quantile. Since $\boldsymbol{Q}$ is orthogonal, the determinant simplifies to $\det(\boldsymbol{\Sigma}) = \det(\boldsymbol{\Lambda})$. The volume of the quantization hypercube, $V_{S_X}$, is determined by the range of the distribution along each principal axis. The extremal points of the ellipsoid are closely correspond to the maximum and minimum along these axes. We denote the eigenvalues of the principal axes corresponding to the points with the maximum and minimum coordinate values as $\lambda_{\max}$ and $\lambda_{\min}$, respectively. After transforming these points back to the original space, the maximum and minimum coordinate values can be represented as:

$$\boldsymbol{v}_{\max}^{\mathrm{org}} = \sqrt{\chi_d^2(\alpha) \cdot \lambda_{\max}} \cdot \boldsymbol{q}_{\max} + \boldsymbol{\mu} \tag{3}$$

$$\boldsymbol{v}_{min}^{\mathrm{org}} = -\sqrt{\chi_d^2(\alpha) \cdot \lambda_{\min}} \cdot \boldsymbol{q}_{\min} + \boldsymbol{\mu} \tag{4}$$

$$V_{S_X} = (\max(\boldsymbol{v}_{\max}^{\mathrm{org}}) - \min(\boldsymbol{v}_{\min}^{\mathrm{org}}))^d \tag{5}$$

where $\boldsymbol{q}_{\max}$ and $\boldsymbol{q}_{\min}$ denote the eigenvectors corresponding to $\lambda_{\max}$ and $\lambda_{\min}$, respectively. Thus, the QSUR becomes:

$$\mathrm{QSUR}_X = \frac{V_{\boldsymbol{X}}}{V_{S_{\boldsymbol{X}}}} = \frac{\frac{\pi^{d/2}}{\Gamma(d/2+1)} \cdot \left(\chi_d^2(\alpha)\right)^{d/2} \cdot \sqrt{\det(\boldsymbol{\Lambda})}}{\left(\max(\sqrt{\chi_d^2(\alpha) \cdot \lambda_{\max}} \cdot |\boldsymbol{q}_{\max}| + \boldsymbol{\mu}) - \min(\sqrt{\chi_d^2(\alpha) \cdot \lambda_{\min}} \cdot |\boldsymbol{q}_{\min}| + \boldsymbol{\mu})\right)^d} \tag{6}$$

Since the magnitude of the mean vector is often smaller than the largest eigenvalue. we neglect the mean vector $\boldsymbol{\mu}$, so $\lambda_{\max} = \lambda_{\min} = \lambda_1$, resulting in:

$$\mathrm{QSUR}_{\boldsymbol{X}} = \frac{\frac{\pi^{d/2}}{\Gamma(d/2+1)} \cdot \left(\chi_d^2(\alpha)\right)^{d/2} \cdot \sqrt{\det(\Lambda)}}{2^d \left(\max(\sqrt{\chi_d^2(\alpha) \cdot \lambda_1} \cdot \boldsymbol{q}_1)\right)^d} = \frac{\frac{\pi^{d/2}}{\Gamma(d/2+1)} \cdot \sqrt{\prod_{i=1}^d \lambda_i}}{2^d \left(\max(\sqrt{\lambda_1} \cdot \boldsymbol{q}_1)\right)^d} \tag{7}$$

From Eq 7, we observe the following: 1) QSUR is proportional to the product of the ratios of each eigenvalue $\lambda_i$ to the largest eigenvalue $\lambda_1$; 2) The maximum component of the eigenvector $\boldsymbol{q}_1$ is inversely proportional to QSUR. As demonstrated in Appendix A.2.1, when the components of $\boldsymbol{q}_1$ take values of $\pm d^{-1/2}$, the denominator in Eq 7 is minimized.

**Influence of linear transformation on QSUR.** Applying a linear transformation $\boldsymbol{T}$ to $\boldsymbol{X} \sim \mathcal{N}(\boldsymbol{\mu}, \boldsymbol{\Sigma})$ results in a transformed distribution $\hat{D} \sim \mathcal{N}(\hat{\boldsymbol{\mu}}, \hat{\boldsymbol{\Sigma}})$, where $\hat{\boldsymbol{\mu}} = \boldsymbol{T}\boldsymbol{\mu}$ and $\hat{\boldsymbol{\Sigma}} = \boldsymbol{T}\boldsymbol{Q}\boldsymbol{\Lambda}\boldsymbol{Q}^{\top}\boldsymbol{T}^{\top}$. Smoothing-based approaches (Xiao et al., 2022; Shao et al., 2023) treat $\boldsymbol{T}$ as a diagonal matrix that scales variances across different channel axes, indirectly reducing the disparities among the eigenvalues $\lambda_i$. However, these methods are particularly sensitive to outliers and uneven mean values, especially when the mean vector $\boldsymbol{\mu}$ contains significant variations like Fig 1(b). Moreover, when quantizing both weights and activations simultaneously, these methods often fail to strike a balance. Rotation-based methods, such as those proposed in (Ashkboos et al., 2024; Liu et al., 2024), reduce outliers in both weights and activations through rotation, thereby decreasing the hypercube volume to increase QSUR. As proven in Appendix A.2.2, this ability to reduce outliers stems from the capacity to modify the matrix $\boldsymbol{Q}$, which improves with increasing dimensionality. When the orthogonal matrix is $\boldsymbol{T} = d^{-\frac{1}{2}}\boldsymbol{H}\boldsymbol{Q}^{\top}$, where $d$ is the dimensionality, and $\boldsymbol{H}$ is a matrix composed of $\pm 1$ entries, the best outlier reduction capability can be achieved.

In combination with Eq7, the maximum QSUR is achieved when:

$$\boldsymbol{T} = c \cdot \boldsymbol{\Lambda}^{-\frac{1}{2}}\boldsymbol{Q}^{\top} \tag{8}$$

where $c$ is an arbitrary scalar. At this point, the maximum utilization rate is given by $\mathrm{QSUR}'' = \frac{\frac{\pi^{d/2}}{\Gamma(d/2+1)}}{2^d}$. Further details can be found in Appendix A.2.3.

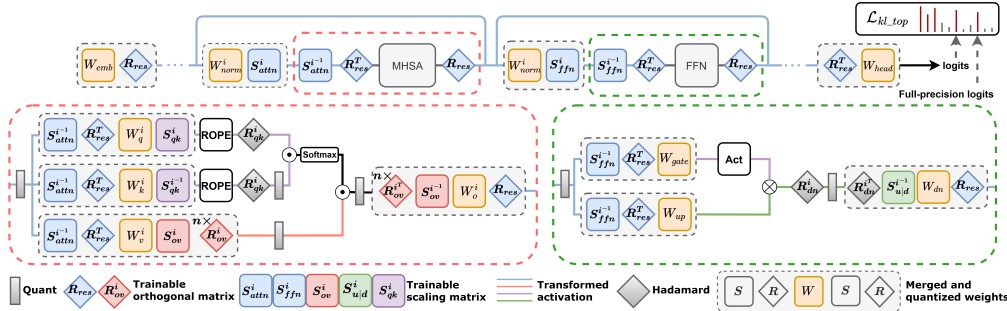

Figure 5: The overall flow diagram of OSTQuant. The top section illustrates how the global orthogonal transformation, $\boldsymbol{R}_{res}$, along with the two scaling transformations, $\boldsymbol{S}_{attn}$ and $\boldsymbol{S}_{ffn}$, collaborate within each block to adjust the distributions across the entire network while maintaining computational invariance. The bottom section highlights four equivalent transformation pairs applied to the FFN and Self-Attention layers. Each fully-connected layer's activation and weight are influenced by one or more of these transformation pairs. During runtime, these transformation pairs are fused with the weights, ensuring minimal runtime overhead.

## 4 METHODOLOGY

### 4.1 ORTHOGONAL AND SCALING TRANSFORMATION-BASED QUANTIZATION

We propose Orthogonal and Scaling Transformation-based Quantization (OSTQuant), a novel framework designed to optimize the distributions of weights and activations in LLMs through learnable equivalent transformation pairs, with the goal of improving quantization performance. The core motivation of OSTQuant is that the combination of orthogonal and scaling transformations enhances the QSUR, as illustrated in Fig 1 and explained in Sec 3.

As illustrated in Fig. 5, OSTQuant applies multiple transformation pairs globally within and across blocks of LLMs. Specifically, four equivalent transformation pairs are learned within each block, with each pair consisting of a learnable diagonal scaling matrix and a learnable orthogonal matrix. These transformations work together to reshape the distributions of weights and activations, making them more quantization-friendly. OSTQuant preserves equivalent transformations at a global network level. As a result, the final output of the network remains unchanged when quantization is not applied, effectively preventing overfitting.

**Equivalent Transformation Pair** We define a transformation pair as $\boldsymbol{T} = \boldsymbol{\Lambda O}$, where $\boldsymbol{T}$ consists of a diagonal scaling matrix $\boldsymbol{\Lambda}$ and a unit orthogonal matrix $\boldsymbol{O}$. As a result, the forward inference process is reformulated as follows:

$$y = \mathcal{Q}(x\boldsymbol{W_1}\boldsymbol{O}\boldsymbol{\Lambda})\mathcal{Q}(\boldsymbol{\Lambda}^{-1}\boldsymbol{O}^T\boldsymbol{W_2}) \tag{9}$$

where $\mathcal{Q}(\cdot)$ represents the quantization operation. Since $\boldsymbol{\Lambda}$ is a diagonal matrix, its inverse is simply the reciprocal of its diagonal elements. We directly optimize $\boldsymbol{O}$ because any orthogonal matrix $\boldsymbol{O}$ can be decomposed into a Hadamard transform and another orthogonal matrix.

Equivalent Transformation Pair has three advantages: **1.** Earnability and Computational Efficiency: Both $\boldsymbol{O}$ and $\boldsymbol{\Lambda}$ are learnable parameters. The inversion of the diagonal matrix $\boldsymbol{\Lambda}$ is computationally simple, enabling efficient forward passes. The orthogonal matrix $\boldsymbol{O}$ can be optimized using gradient-based optimizers, such as RiemannAdam (Bécigneul & Ganea, 2018), which supports optimization on Stiefel Manifolds. This allows the entire process to fully leverage first-order gradient information for end-to-end learning. **2.** Equivalence Preservation: Ignoring the effects of quantization, the forward process remains mathematically equivalent to the original model. This ensures that activations and weights retain their consistency while making their distributions more quantization-friendly, thus reducing the risk of overfitting. **3.** After optimization, $\boldsymbol{O}$ and $\boldsymbol{\Lambda}$ can be directly merged into the existing weights, meaning no additional computational overhead or parameters are introduced during deployment, ensuring efficient inference.

The optimization objective for the entire network can be formalized as:

$$\arg \min_{\boldsymbol{A}_i, \boldsymbol{O}_i} \mathcal{L}(\hat{y}, y; \boldsymbol{A}_i, \boldsymbol{O}_i, \theta) \tag{10}$$

where $\theta$ represents the frozen network parameters, and $\mathcal{L}(\hat{y}, y)$ represents the loss between the quantized network output $\hat{y}$ and the full-precision output $y$.

**Weight Outlier Minimization Initialization (WOMI)** As shown in (Cholakov et al., 2023), Weight typically follow a Gaussian distribution with zero mean. Therefore, we initialize the orthogonal matrix $O$ using the Eq 27 provided in appendix. For the global orthogonal matrix $R_{res}$, we concatenate the weights of all linear layers receiving residual inputs along the input channels, denoted as $W^{n \cdot oc \times ic}$, and perform eigenvalue decomposition on its covariance to obtain the eigenmatrix $Q_W$. Then, we initialize $R_{res} = (Q_W)H^T$, $H$ is normalized Hadamard matrix. For the orthogonal matrices of the Out-projection and Value-projection layers, we split the $O$ matrix along the head dimension and apply the same initialization method as $R_{res}$. For all scaling matrices, we initialize them as identity matrices.

**Inter-Block Learning.** As illustrated in the upper half of Fig 5, the global $R_{res}$ is applied at the embedding layer and propagates through each residual path within the LLM via the projection layers. This transformation rotates activations throughout the entire residual path. Due to the norm-preserving property of unitary orthogonal matrices (Tseng et al., 2024), we can bypass the RMSNorm layers and apply an inverse transformation to the inputs of each residual connection's projection layer and the final output head, ensuring equivalence is maintained.

Additionally, for the two normalization layers in each block and their respective projection layers, we introduce two diagonal scaling matrices, $S_{attn}^i$ and $S_{ffn}^i$, to smooth channel-wise differences. The matrix $R_{res}$ simultaneously rotates activations along the residual paths and adjusts the weights of multiple projection layers. The scaling matrices $S_{attn}^i$ and $S_{ffn}^i$ apply scaling to the outputs of the RMSNorm layers and their corresponding projection layers. These transformations can be absorbed into the corresponding weight matrices: the orthogonal transformation $R_{res}$ merges with the projection weights along the residual paths, and the scaling matrices are incorporated into the weight vectors of the RMSNorm layers. As shown in Fig 5, by fusing $R_{res}$ with all projection weights during optimization, we effectively learn how distribution shifts in weights and activations impact model accuracy. This approach helps mitigate the effects of quantization errors by adjusting for these shifts, thus improving model performance.

**Intra-Block Learning.** As illustrated in the lower half of Fig 5, we introduce two equivalent transformation pairs within the Multi-Head Self-Attention layer of each transformer block. Specifically, the value ($V$) and output ($O$) projection layers are transformed across layers. For each attention head, the data flow is expressed as:

$$Y = P^h \cdot X^h \cdot (W_v^h R_{ov}^h S_{ov}^h) \cdot (S_{ov}^{h^{-1}} R_{ov}^{h^T} W_o^h) \tag{11}$$

Here, $h$ denotes the head index, $P^h$ is the attention matrix, and $X^h$ is the input to head $h$. We learn a rotation transformation $R_{ov}^h$ and a scaling transformation $S_{ov}^h$ for each attention head. These transformations aim to improve QSUR for both the value cache and the output projection layer.

After the Rotary Positional Encoding (ROPE) operation, the output query and key can naturally undergo an equivalent scaling transformation $S_{qk}$, similar to the approach in (Hu et al., 2024). Due to the multiplicative nature of positional encoding, this scaling transformation can be incorporated into the weight matrices $W_q$ and $W_k$. To further enhance the quantization efficiency of the key cache, we apply an additional Hadamard transformation like Quarot (Ashkboos et al., 2024) to the outputs of the query and key. Similar to $S_{qk}$, we can optimize the diagonal matrices in the up-projection and down-projection of the FFN layer. However, the inverse of the Hadamard transformation is fused into $W_{down}$ from the very beginning.

## 4.2 KL-TOP LOSS.

While LLMs are typically trained on vast datasets, OS-TQuant optimization is often performed using a much smaller sample set, typically around 1,000 examples. In this limited-data setting, directly applying original cross-entropy (CE) loss can result in accuracy drop. As shown in Tab 1, despite the quantized model exhibiting lower perplexity compared to its

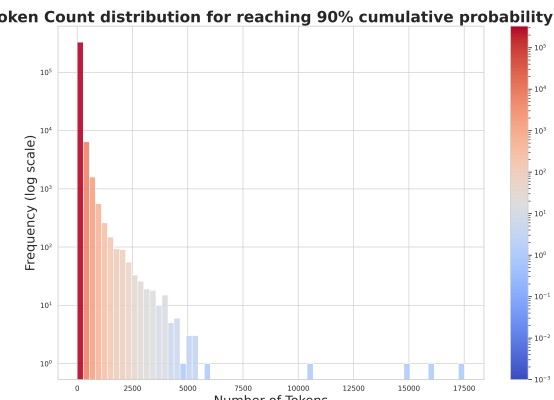

Figure 6: The distribution of the number of tokens required to accumulate 90% of the total prediction probability in the LLaMA-2-7B model.

full-precision counterpart after training with CE loss, its performance on zero-shot tasks declines. One likely explanation is that small and simple datasets, such as WikiText-2 (Merity et al., 2016), may not fully utilize the capacity of LLMs. Consequently, relying solely on CE loss, which focuses on a single label, might cause the model to overfit to a narrow set of features, thereby compromising its emergent capabilities.

Table 1: Impact of different loss on Wiki PPL and Arc (Boratko et al., 2018) accuracy for LLaMA models: While Origin loss reduces PPL, zero-shot scores better reflect the model's performance.

| Model | Loss Type | Wiki PPL | Arc-Easy Score | Arc-Challenge Score |
|---|---|---|---|---|
| LLaMA-2-7B | Origin | **5.38** | 69.87 | 42.41 |
| | KL-Top | 5.94 | **72.69** | **44.62** |
| LLaMA-2 13B | Origin | **5.12** | 75.09 | 46.08 |
| | KL-Top | 5.25 | **75.29** | **47.10** |
| LLaMA-3 8B | Origin | **6.80** | 76.68 | 49.26 |
| | KL-Top | 7.29 | **76.73** | **49.32** |

A natural idea is to align the prediction distributions before and after quantization to reduce overfitting risks, such as using the Kullback-Leibler (KL) divergence for optimization. However, this approach also faces challenges. LLMs typically have vocabularies of tens of thousands or more(e.g., LLaMA-3-8B has over 100,000 tokens). As illustrated in Fig 6, the prediction results of the full-precision model follow a severe long-tail distribution, with only a small number of tokens having significant probabilities. If we directly apply KL divergence over all classes, the loss may be dominated by uninformative classes with negligible probabilities, adding noise to the training process.

To address this, we propose the KL-Top loss function, which computes KL divergence over only the top-$k$ classes with the highest probabilities. By focusing optimization on the model's primary predictions, this approach enhances gradient quality. In the global KL loss, low-probability values can introduce noise, leading to inaccurate gradient updates. By restricting the computation to the top-$k$ classes, the model receives clearer and more informative gradients. Moreover, when dealing with a large number of classes (e.g., over 100,000), both computation and memory costs become substantial. Limiting the calculation to the top-$k$ classes (e.g., $k = 1000$) not only reduces complexity but also accelerates the training process. The KL-Top loss is calculated as follows:

$$idxs = \text{top}k(\boldsymbol{z}) \tag{12}$$

$$\mathcal{L} = \sum_{i \in idxs} \boldsymbol{z}[i] \log \left( \frac{\boldsymbol{z}[i]}{\hat{\boldsymbol{z}}[i]} \right) \tag{13}$$

where $\boldsymbol{z}$ and $\hat{\boldsymbol{z}}$ are the prediction distributions before and after quantization, respectively.

## 5 EXPERIMENTS

**Models and Datasets.** We apply our method to the entire LLaMA family, including LLaMA-1 (7B–30B) (Touvron et al., 2023a), LLaMA-2 (7B–13B) (Touvron et al., 2023b), and LLaMA-3-8B. We report perplexity (PPL) scores on the WikiText2 (Merity et al., 2016) test set. However, as mentioned in Tab 1, perplexity may not fully reflect the model's true performance after quantization, zero-Shot tasks better reflect the model's actual performance. Therefore, we also evaluate the models on up to nine zero-shot tasks using the `lm-evaluation-harness` (version 0.4.4) (Gao et al., 2024), including BoolQ (Clark et al., 2019), HellaSwag (Zellers et al., 2019), LAMBADA (OpenAI) (Radford et al., 2019), OpenBookQA (OBQA) (Mihaylov et al., 2018), PIQA (Bisk et al., 2020), SIQA (Sap et al., 2019), WinoGrande (Sakaguchi et al., 2021), ARC-Easy, and ARC-Challenge (Boratko et al., 2018).

**Baselines and Implementation Details.** In addition to the basic RTN approach, we benchmark our approach against SmoothQuant (Xiao et al., 2022), GPTQ (Frantar et al., 2022), and current state-of-the-art methods such as Quarot (Ashkboos et al., 2024) and SpinQuant (Liu et al., 2024) for both weight-only and weight-activation quantization. All activations are quantized using per-token asymmetric quantization without any pruning operations, while weights are quantized using symmetric per-channel quantization. We use RiemannAdam (Bécigneul & Ganea, 2018) to optimize all unit orthogonal matrices and scaling matrices. During the distribution optimization phase, we use 1,000 samples from WikiText2, each with a token length of 2,048, and iterate 150 times with a batch size of 8. We apply cosine learning rate decay, setting the initial learning rate for all orthogonal matrix parameters to $2 \times 10^{-2}$ and for scaling parameters to $3 \times 10^{-2}$.

## 5.1 OVERALL RESULTS

Table 2: Comparison of perplexity on WikiText2 and averaged accuracy on nine Zero-Shot tasks. Results for SmoothQuant, GPTQ, OmniQuant, AWQ, and QuaRot are based on official code and SpinQuant's results for LLaMA-2/3 using official weights, with LLaMA-1 from the official code.

| #Bits W-A-KV | Method | LLaMA-3 8B 0-shot⁹ Avg.(↑) | Wiki (↓) | LLaMA-3 70B 0-shot⁹ Avg.(↑) | Wiki (↓) | LLaMA-2 7B 0-shot⁹ Avg.(↑) | Wiki (↓) | LLaMA-2 13B 0-shot⁹ Avg.(↑) | Wiki (↓) | LLaMA-2 70B 0-shot⁹ Avg.(↑) | Wiki (↓) | LLaMA 7B 0-shot⁹ Avg.(↑) | Wiki (↓) | LLaMA 13B 0-shot⁹ Avg.(↑) | Wiki (↓) | LLaMA 30B 0-shot⁹ Avg.(↑) | Wiki (↓) |
|---|---|---|---|---|---|---|---|---|---|---|---|---|---|---|---|---|---|
| 16-16-16 | FloatingPoint | 68.09 | 6.14 | 73.81 | 2.86 | 65.21 | 5.47 | 67.61 | 4.88 | 71.59 | 3.32 | 64.48 | 5.68 | 66.67 | 5.09 | 70.00 | 4.10 |
| 4-16-16 | RTN | 63.70 | 8.13 | 31.15 | 1e5 | 61.27 | 7.02 | 60.24 | 6.39 | 69.62 | 3.87 | 62.67 | 7.94 | 63.45 | 8.60 | 65.69 | 6.13 |
| | SmoothQuant | 62.79 | 8.12 | 67.94 | 6.70 | 58.88 | 8.03 | 62.03 | 5.86 | 65.93 | 5.50 | 62.24 | 7.46 | 62.69 | 18.75 | 65.69 | 5.80 |
| | GPTQ | 61.03 | 7.43 | 31.45 | 9e3 | 60.86 | 9.84 | 64.71 | 5.79 | 70.96 | 3.94 | 60.15 | 7.93 | 64.36 | 6.58 | 66.95 | 5.26 |
| | Omniquant | 65.66 | 7.19 | - | - | 63.19 | 5.74 | 66.38 | 5.02 | 71.04 | 3.47 | 63.42 | 5.86 | 66.22 | 5.21 | 69.07 | 4.25 |
| | AWQ | 67.03 | 7.36 | 68.92 | 5.92 | 63.89 | 5.83 | 66.25 | 5.07 | 70.88 | 4.03 | 63.30 | 5.97 | 65.58 | 5.28 | 69.44 | 4.28 |
| | QuaRot | 67.27 | 6.53 | 72.93 | 3.53 | 64.30 | 5.62 | 66.95 | 5.00 | 71.21 | 3.41 | 63.40 | 5.83 | 65.91 | 5.20 | 69.73 | 4.27 |
| | SpinQuant | 66.54 | **6.49** | 72.90 | 3.49 | 63.59 | **5.58** | 67.14 | 5.00 | 71.12 | 3.43 | 63.94 | **5.76** | 66.32 | **5.16** | 69.62 | 4.21 |
| | **OSTQuant** | **67.80** | 6.53 | **73.69** | **3.19** | **64.37** | 5.64 | **67.31** | **4.94** | **71.48** | **3.41** | **64.13** | 5.81 | **66.62** | 5.21 | **69.84** | **4.19** |
| 4-4-16 | RTN | 33.42 | 6e2 | 31.21 | 8e3 | 32.44 | nan | 30.86 | 8e3 | 30.90 | 7e4 | 32.51 | 7e3 | 31.63 | 3e4 | 31.57 | 2e3 |
| | SmoothQuant | 33.04 | 1e3 | 34.67 | 2e2 | 32.13 | nan | 34.26 | 1e3 | 35.86 | 3e2 | 34.42 | 3e2 | 33.29 | 6e2 | 34.64 | 1e3 |
| | GPTQ | 32.98 | 5e2 | 31.47 | 4e4 | 32.72 | nan | 30.11 | 4e3 | 30.86 | nan | 32.12 | 1e3 | 31.51 | 3e3 | 30.88 | 2e3 |
| | QuaRot | 61.69 | 8.02 | 65.56 | 6.35 | 61.87 | 6.05 | 65.13 | 5.35 | 69.96 | 3.78 | 61.76 | 6.22 | 64.46 | 5.50 | 68.14 | 4.57 |
| | SpinQuant | 64.11 | 7.28 | 66.99 | 6.10 | 57.37 | 6.78 | 63.23 | 5.24 | 70.58 | 3.68 | 61.82 | 6.08 | 64.59 | **5.36** | 68.08 | 4.53 |
| | **OSTQuant** | **65.14** | **7.24** | **72.21** | **3.97** | **63.90** | **5.60** | **66.24** | **5.14** | **70.92** | **3.57** | **62.72** | **6.04** | **65.80** | 5.40 | **68.52** | **4.43** |
| 4-4-4 | RTN | 33.18 | 7e2 | 30.82 | 8e3 | 32.67 | nan | 30.93 | 7e3 | 31.73 | 7e4 | 32.87 | 1e4 | 31.33 | 3e4 | 31.64 | 2e3 |
| | SmoothQuant | 32.96 | 1e3 | 33.76 | 3e2 | 32.12 | nan | 33.36 | 1e3 | 35.54 | 3e2 | 33.32 | 3e2 | 33.28 | 5e2 | 34.65 | 1e3 |
| | GPTQ | 33.71 | 6e2 | 31.20 | 4e4 | 33.52 | nan | 27.85 | 5e3 | 31.09 | nan | 31.80 | 2e3 | 30.63 | 3e3 | 31.07 | 2e3 |
| | Omniquant | 32.33 | 4e2 | - | - | 48.40 | 14.26 | 50.35 | 12.30 | - | - | 48.46 | 11.26 | 45.63 | 10.87 | 45.04 | 12.35 |
| | QuaRot | 61.38 | 8.18 | 65.33 | 6.6 | 61.48 | 6.11 | 65.16 | 5.39 | 70.30 | 3.80 | 61.22 | 6.26 | 64.59 | 5.53 | 68.08 | 4.60 |
| | SpinQuant | 64.10 | 7.35 | 66.31 | 6.24 | 62.01 | 5.96 | 64.13 | 5.74 | 70.57 | 3.61 | 61.32 | 6.12 | 64.95 | **5.39** | 68.14 | 4.55 |
| | **OSTQuant** | **65.37** | **7.29** | **71.69** | **4.01** | **63.18** | **5.91** | **65.41** | **5.25** | **70.84** | **3.59** | **62.55** | **6.07** | **65.43** | 5.40 | **68.20** | **4.42** |

**Quantization performance.** As shown in Tab 2, our method consistently outperforms previous SOTA approaches across almost all configurations and models. Under the 4-16-16 setup, OSTQuant surpasses all prior methods, maintaining at least 99.5% floating-point (FP) accuracy in zero-shot tasks. Compared to other weight-only methods like GPTQ and AWQ, OSTQuant further narrows the gap with FP models. In the most challenging LLaMA-3-8B model, OSTQuant achieves only a 0.29-point performance drop in zero-shot evaluations, whereas other methods incur losses exceeding 1.55 points. Even in the highly challenging 4-4-4 setting, our approach retains a significant performance gain, outperforming the SOTA method, SpinQuant, by around 1 point across multiple models. Notably, when the KV cache is not quantized (in the 4-4-16 setup), OSTQuant achieves a significant performance boost over SpinQuant, with gains up to 6.53 points (LLaMA-2 7B). These substantial performance improvements demonstrate the effectiveness of our approach. More detailed results can be seen in Appendix A.6. Once activation is quantized, rotation-based methods significantly outperform smooth-based methods, confirming that latter struggle with outliers and uneven distributions. In Fig 3, the QSUR across different methods show a clear positive correlation with model performance. Our approach achieves the highest QSUR, effectively mitigating the challenges of outliers and uneven distributions that hinder prior methods, leading to improved model accuracy.

Table 3: The speedup and memory saving factor of LLaMA models with different parameter sizes and sequence lengths, compared between our 4-bit implementation and FP16. All tests were conducted on a Transformer block with batch size 4 on a 3090 GPU.

| Model Size | Prefill Speedup (Seqlen) | | | | | | Memory Saving Factor (Seqlen) | | | | | |
|---|---|---|---|---|---|---|---|---|---|---|---|---|
| | 256 | 512 | 1024 | 2048 | 4096 | 8192 | 256 | 512 | 1024 | 2048 | 4096 | 8192 |
| 7B | 2.24x | 2.27x | 2.23x | 2.14x | 2.11x | 2.02x | 3.48x | 3.34x | 3.12x | 2.86x | 2.57x | 2.34x |
| 8B | 2.42x | 2.52x | 2.52x | 2.43x | 2.36x | 2.23x | 3.48x | 3.36x | 3.12x | 2.77x | 2.38x | 2.00x |
| 13B | 2.62x | 2.68x | 2.63x | 2.52x | 2.83x | 2.32x | 3.64x | 3.51x | 3.30x | 3.02x | 2.70x | 2.43x |
| 30B | 3.18x | 3.01x | 2.98x | 3.40x | 2.84x | 2.68x | 3.70x | 3.59x | 3.42x | 3.15x | 2.83x | 2.53x |

**Speedup and memory savings.** OSTQuant incurs only negligible loss in 4-bit quantization, making 4-bit inference feasible. As shown in Tab 3, OSTQuant delivers an average inference speedup of over $2\times$ and memory savings exceeding $3.5\times$, demonstrating the significant improvements in inference efficiency. Detailed speedup and memory saving results can refer to Tab 9. Moreover, OSTQuant provides substantial advantages in training speed compared to block reconstruction-based methods. With only 150 iterations and a minimal number of learnable parameters, we optimize the 7B and 13B models in around 20 minutes, and the 30B model in 120 minutes, achieving up to $5.3\times$ speedups compared to OmniQuant, refer to Tab 11 for more details.

## 5.2 ABLATION STUDY

**Effect of different transformation.** We ablate the effects of various transformation matrices on LLaMA-2 7B, identifying four groups where orthogonal and scaling equivalent transformations can

Table 4: Ablation study on the impact of different transformation matrices on Wiki PPL and zero-shot[9] score for LLaMA-2 7B under W4A4KV4 quantization.

| Metric | Baseline | $+R_{res}$ | $+S_{res}$ | $+R_{dn}$ | $+S_{u\|d}$ | $+R_{qk}$ | $+S_{qk}$ | $+R_{ov}$ | $+S_{ov}$ |
|---|---|---|---|---|---|---|---|---|---|
| Wiki PPL | nan | 9.70 | 9.46 | 6.16 | 6.00 | 5.92 | 5.92 | 5.94 | **5.91** |
| Zero-shot[9] | 33.51 | 54.33 | 53.74 | 61.75 | 61.79 | 62.35 | 62.56 | 63.11 | **63.18** |

be applied. Tab 4 presents the contribution of each parameter group under W4A4KV4 setup. Our results show that the global orthogonal transformation $R_{res}$ brings the largest improvement, followed closely by $R_{down}$. Notably, scaling transformations $S$ further build on the orthogonal transformations $R$ by effectively balancing variance across channels, thereby minimizing quantization losses and enhancing model performance.

Table 5: Effect of different optimizers on zero-shot[9] performance of LLaMA models under the W4A4KV4 configuration. LR1 and LR2 represent the learning rates for the scaling matrices and unitary orthogonal matrices.

| Model | Optimizer Type | Best Steps | Best LR1 | Best LR2 | Zero-Shot[9] Score |
|---|---|---|---|---|---|
| | Cayley SGD | 150 | 1.50 | 0.20 | 63.11 |
| LLaMA-2-7B | Riemann SGD | 500 | 0.10 | 0.02 | 63.09 |
| | Riemann Adam | 150 | 0.02 | 1e-3 | **63.18** |
| | Cayley SGD | 200 | 1.50 | 0.2 | 64.77 |
| LLaMA-2-13B | Riemann SGD | 500 | 0.1 | 0.02 | 65.19 |
| | Riemann Adam | 150 | 0.02 | 0.002 | **65.41** |

**Different Manifold Optimizers.** Since the unit orthogonal matrix resides on a Stiefel manifold, we explore various manifold optimizers to optimize it, including CayleySGD (Li et al., 2020), RiemannSGD, and RiemannAdam (Bécigneul & Ganea, 2018). Tab 5 compares these methods and shows that CayleySGD typically requires a higher learning rate to perform well, RiemannSGD needs more iterations, while RiemannAdam delivers the best results with the fewest iterations. We also discover that using a learning rate for the Stiefel manifold 10 times larger than that for scaling transformation parameters leads to better results.

Table 6: Ablation study of k values on Zero-Shot[9] score and Wiki PPL for W3-only and W4A4KV4 configurations of LLaMA-2 7B.

| Setting | Metric | $k$=5 | $k$=50 | $k$=100 | $k$=500 | $k$=1000 | $k$=5000 | $k$=10000 |
|---|---|---|---|---|---|---|---|---|
| W3 Only | Zero-Shot[9] Score | 61.87 | 61.88 | 61.75 | 62.18 | **62.30** | 61.25 | 61.21 |
| | Wiki PPL | 6.06 | 6.116 | 6.13 | 6.07 | **6.06** | 6.06 | 6.12 |
| W4A4KV4 | Zero-Shot[9] Score | 62.4 | 62.13 | 62.38 | 62.34 | **63.18** | 62.44 | 62.11 |
| | Wiki PPL | 5.99 | 5.96 | 5.95 | 5.96 | 5.96 | **5.93** | 5.94 |

**Influence of k in KL-Top loss.** The parameter $k$ in Eq. 12 defines the number of classes considered when calculating the KL-Top loss, balancing optimization difficulty with semantic richness. Both excessively large or small $k$ values negatively impact optimization. Tab 6 shows a comparison of different $k$ values. Furthermore, we analyze whether to apply softmax before or after the top-$k$ selection. Our experiments indicate that setting $k$ to 1,000 processing produces the best outcomes.

## 6 CONCLUSION

In this paper, we introduce OSTQuant, a novel post-training quantization method designed to enhance the efficiency of large language models (LLMs). Central to OSTQuant is the Quantization Space Utilization Rate (QSUR), a new metric we proposed to effectively assess the quantizability of transformed data by measuring its space utilization within the quantization space. Complemented by mathematical derivations, QSUR provides theoretical guidance for optimizing single data distributions across the entire quantization space. Leveraging this insight, OSTQuant employs a learnable equivalent transformation pair composed of orthogonal and scaling transformations to optimize the distributions of weights and activations. Additionally, we introduce the KL-Top loss function to mitigate noise during optimization while retaining richer semantic information, even with the limited calibration data typically available in PTQ. Extensive experiments on various LLMs and benchmarks demonstrate that OSTQuant outperforms existing quantization methods. These results highlight the effectiveness of optimizing data distributions across the quantization space and underscore OSTQuant's potential to advance LLM quantization, making these models more efficient and practical for deployment in resource-constrained environments.

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

## A APPENDIX

### A.1 QUANTIZATION PRELIMINARIES

**Quantization & Dequantization.** Quantization typically refers to mapping a floating-point number to a discrete interval with integer number. Here, we only consider uniform quantization. The quantization process can be expressed as follows:

$$\boldsymbol{X}^I = \text{clamp}\left(\left\lfloor \frac{\boldsymbol{X}}{s} \right\rceil + zp^I, 0, 2^{n^I} - 1\right) \tag{14}$$

$$s = \frac{x_{\max} - x_{\min}}{2^{n^I} - 1} \tag{15}$$

$$zp^I = \left\lfloor \frac{-x_{\min}}{s} \right\rceil \tag{16}$$

$$\boldsymbol{X}' = (\boldsymbol{X}^I - zp^I) \cdot s \tag{17}$$

where $\boldsymbol{X}$ is the floating-point tensor, $\boldsymbol{X}^I$ is its quantized counterpart and $\boldsymbol{X}'$ is the dequantized resuls of $\boldsymbol{X}^I$. Here, $n^I$ represents the number of bits (e.g., 8). $s$ is the quantization step size, determined by $x_{\min}, x_{\max}$, and $n^I$. clamp represents truncation function. The choice of $s$ greatly affects the accuracy of the quantized model. We can obtain $s$ from the activations of some samples, which is called static quantization. Alternatively, we can derive it from runtime statistics, known as dynamic quantization. Quantization can also be distinguished by its granularity into per-channel quantization and per-token quantization (Yao et al., 2022).

### A.2 SOME DEDUCTIONS ABOUT QSUR

#### A.2.1 THE COORDINATES OF THE EXTREMUM POINT AND THE INFLUENCE OF ROTATION MATRIX

From Eq6:

$$\text{QSUR}_X = \frac{V_X}{V_{S_X}} = \frac{\frac{\pi^{d/2}}{\Gamma(d/2+1)} \cdot \left(\chi_d^2(\alpha)\right)^{d/2} \cdot \sqrt{\det(\boldsymbol{\Lambda})}}{\left(\max(\sqrt{\chi_d^2(\alpha) \cdot \lambda_{\max}} \cdot |\boldsymbol{q}_{\max}| + \boldsymbol{\mu}) - \min(\sqrt{\chi_d^2(\alpha) \cdot \lambda_{\min}} \cdot |\boldsymbol{q}_{\min}| + \boldsymbol{\mu})\right)^d}$$

Given that the eigenvalues and the mean vector are fixed, the volume of the hypercube is determined by the eigenmatrix $\boldsymbol{Q}$. For any eigenvector $qi \in \boldsymbol{Q}$, we have the condition:

$$qi1^2 + q_{i2}^2 + \cdots + q_{id}^2 = 1 \tag{18}$$

Let $q_{i\max} = \max(q_{i1}, q_{i2}, \ldots, q_{id})$, which implies that for any $i$, we have $q_{ij} \leq q_{i\max}$. Therefore, it follows that:

$$q_{i1}^2 + q_{i2}^2 + \cdots + q_{id}^2 \leq d \cdot q_{i\max}^2 \tag{19}$$

This yields the inequality:

$$1 \leq d \cdot q_{i\max}^2 \tag{20}$$

$$|q_{i\max}| \geq d^{\frac{1}{2}} \tag{21}$$

Consequently, when all elements in $\boldsymbol{Q}$ are $\pm d^{-\frac{1}{2}}$, the quantization range is minimized. Under these conditions, the QSUR' can be expressed as:

$$\text{QSUR}' = \frac{\frac{\pi^{d/2}}{\Gamma(d/2+1)} \cdot \left(\chi_d^2(\alpha)\right)^{d/2} \cdot \sqrt{\det(\boldsymbol{\Lambda})}}{\left(\max\left(\sqrt{\chi_d^2(\alpha) \cdot \lambda_{\max}} \cdot d^{-\frac{1}{2}} + \boldsymbol{\mu}\right) - \min\left(-\sqrt{\chi_d^2(\alpha) \cdot \lambda_{\min}} \cdot d^{-\frac{1}{2}} + \boldsymbol{\mu}\right)\right)^d} \tag{22}$$

When the mean vector $\boldsymbol{\mu} = 0$, the QSUR' can be further simplified to:

$$\text{QSUR}' = \frac{\frac{\pi^{d/2}}{\Gamma(d/2+1)} \cdot \sqrt{\prod_{i=1}^d \lambda_i}}{2^d \cdot \left(\sqrt{\lambda_1} \cdot d^{-\frac{1}{2}}\right)^d} \tag{23}$$

### A.2.2 THE BEST ORTHOGONAL MATRIX

In the linear transformation paragraph 3 of Sec3 , we observe that applying a linear transformation $\boldsymbol{T}$ to $X \sim \mathcal{N}(\boldsymbol{\mu}, \boldsymbol{\Sigma})$ results in a transformed distribution:

$$\hat{X} \sim \mathcal{N}(\hat{\boldsymbol{\mu}}, \hat{\boldsymbol{\Sigma}}) \tag{24}$$

Since the transformation $\boldsymbol{TQ}$ remains an orthonormal matrix after applying the unitary transformation, the transformed eigenvectors are given by:

$$\mathbf{TQ} = [\mathbf{q_1}, \mathbf{q_2}, \dots, \mathbf{q_d}] \tag{25}$$

Movited by Eq 21, we aim for the following transformation:

$$\boldsymbol{TQ} = d^{-\frac{1}{2}} \boldsymbol{H} \tag{26}$$

where $d$ is the dimensionality, and $\boldsymbol{H}$ is a matrix composed of $\pm 1$ entries. Solving for $\boldsymbol{T}$ yields:

$$\boldsymbol{T} = d^{-\frac{1}{2}} \boldsymbol{H} \boldsymbol{Q}^{\top} \tag{27}$$

The covariance matrix after transformation becomes:

$$\hat{\boldsymbol{\Sigma}} = d^{-\frac{1}{2}} \cdot \boldsymbol{H} \boldsymbol{Q}^{\top} \boldsymbol{Q} \boldsymbol{\Lambda} \boldsymbol{Q}^{\top} \boldsymbol{Q} \boldsymbol{H}^{\top} \cdot d^{-\frac{1}{2}} \tag{28}$$

$$= d^{-1} \boldsymbol{H} \boldsymbol{\Lambda} \boldsymbol{H}^{\top} \tag{29}$$

### A.2.3 THE BEST TRANSFORM MATRIX

Inspired by Sec 3, we know that applying a linear transformation $\boldsymbol{T}$ to $X \sim \mathcal{N}(\boldsymbol{\mu}, \boldsymbol{\Sigma})$ results in a transformed $\hat{X} \sim \mathcal{N}(\hat{\boldsymbol{\mu}}, \hat{\boldsymbol{\Sigma}})$. Ignoring the mean vector, we present a transformation matrix, composed of a diagonal scaling transformation and an orthonormal transformation, which minimizes the QSUR:

$$\boldsymbol{T} = \boldsymbol{\Lambda}^{-\frac{1}{2}} \boldsymbol{Q}^{\top} \tag{30}$$

Under this transformation, the covariance matrix becomes:

$$\hat{\boldsymbol{\Sigma}} = \boldsymbol{\Lambda}^{-\frac{1}{2}} \boldsymbol{Q}^{\top} \boldsymbol{Q} \boldsymbol{\Lambda} \boldsymbol{Q}^{\top} \boldsymbol{Q} \boldsymbol{\Lambda}^{-\frac{1}{2}} \tag{31}$$

$$= \boldsymbol{I} \tag{32}$$

By substituting this result into Eq 7, we achieve the maximum QSUR:

$$\text{QSUR}'' = \frac{\frac{\pi^{d/2}}{\Gamma(d/2+1)}}{2^d} \tag{33}$$

## A.3 ADDITIONAL ABLATION EXPERIMENTS

### A.3.1 THE EFFECT OF WEIGHT OUTLIER MINIMIZATION INITIALIZATION

Weight Outlier Minimization Initialization(WOMI) is used to initialize trainable orthogonal matrices. This approach not only reduces outliers in the weights but also leverages the properties of Hadamard matrices to mitigate inter-channel disparities in activations, thereby improving the initial QSUR for both weights and activations.

We visualized the impact of WOMI on the weights. As shown in Fig 7, the original weight distribution exhibits significant variations across input and output channels. While QuaRot reduces inter-channel differences, noticeable spikes remain. WOMI, by leveraging the Hadamard matrix and the covariance matrix of the weight distribution, further smooths these inter-channel differences, effectively reducing the quantization space and relative quantization error. Additional visual results for other layers are presented in Fig 13.

We conducted additional experiments to investigate the impact of WOMI on the performance of quantized models. Tab 7 presents the performance of LLaMA-2-7B and LLaMA-3-8B models initialized with WOMI and random Hadamard matrices. WOMI achieves lower perplexity and higher few-shot accuracy under both W4A4KV4 and W4A16KV16 configurations, showcasing its effectiveness. Interestingly, WOMI demonstrates greater performance improvements in W4-only quantization settings compared to W4A4KV4. This is likely due to WOMI's superior capability in minimizing weight quantization errors, which is especially critical in W4-only configurations.

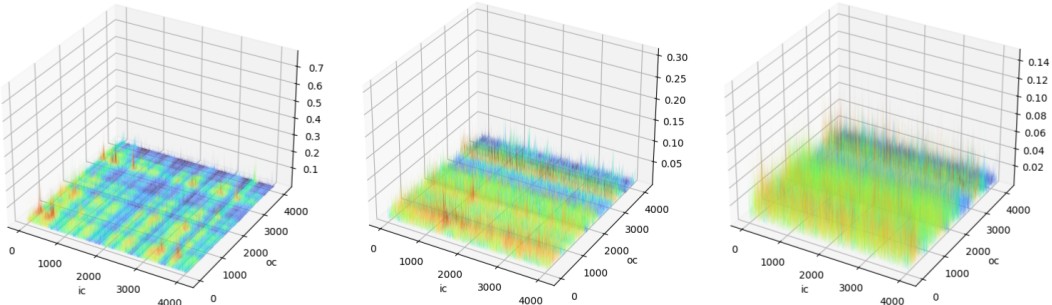

(a) Original weight distribution   (b) Hadamard. Relative L1 Error: 0.79   (c) WOMI. Relative L1 Error: 0.42

Figure 7: Impact of WOMI transform and Hadamard transform on LLaMA-2-7B weight (weight of Query projection in Layer 0) quantization. The relative error is computed by dividing the mean absolute error (MAE) between the original and quantized weights by the mean absolute value of the original weights.

Table 7: Comparison of the impact of WOMI initialization and Hadamard initialization on the performance of quantized models.

| Model | Quant Setting | Method | Zero-Shot[9] | Wiki PPL |
|-------|---------------|--------|-----------|----------|
| LLaMA-2-7B | Full-Precision | - | 65.21 | 5.47 |
| | W4A16KV16 | Hadamard | 63.32 | 5.62 |
| | W4A16KV16 | WOMI | 63.45 | 5.59 |
| | W4A4KV4 | Hadamard | 61.47 | 6.11 |
| | W4A4KV4 | WOMI | 61.52 | 6.09 |
| LLaMA-3-8B | Full-Precision | - | 68.09 | 6.14 |
| | W4A16KV16 | Hadamard | 67.27 | 6.53 |
| | W4A16KV16 | WOMI | 67.41 | 6.48 |
| | W4A4KV4 | Hadamard | 61.38 | 8.18 |
| | W4A4KV4 | WOMI | 61.40 | 8.17 |

### A.3.2 THE EFFECT OF KL-TOP LOSS

As shown in Tab 8 , the results indicate that using SpinQuant(Liu et al., 2024) alone, even with the introduction of the KL-Top loss function, does not lead to significant performance improvements and may even cause some degradation. However, when combined with orthogonal and scaling transformation pairs, the quantization performance improves significantly. For OSTQuant, using CE loss results in overfitting on the calibration set, and this issue is alleviated by the introduction of the KL-Top loss function.

### A.4 INFERENCE EFFICIENCY AND QUANTIZATION OVERHEAD

Tab 9 shows the prefill time and memory usage of LLaMA models with different parameter sizes and sequence lengths, compared between our 4-bit implementation and FP16. The inference environment features an Intel(R) Xeon(R) Gold 5317 CPU and an Nvidia 3090 GPU. The 4-bit matrix multiplication kernel was implemented using cutlass of nvidia, while the self-attention mechanism was realized with PyTorch's native SDPA (scaled dot product attention) function. All tests were conducted 500 times, with the median value taken as the final result. Benefiting from efficient low-precision computation units within CUDA cores and reduced access overhead, OSTQuant achieves over 2× speedup across various model sizes, and approximately 3× acceleration on the challenging LLaMA-30B model.

Fig 8 displays the performance and model size of various LLaMA models under different quantization bitwidths. Each line represents a different model (LLaMA-2-7B, LLaMA-3-8B, LLaMA-2-13B, LLaMA-30B, and LLaMA-3-70B) with quantization bitwidths ranging from 2 bits to 16 bits. As

Table 8: Performance of SpinQuant with the introduction of the KL-Top loss function on 9 zero-shot dataset tasks and the Perplexity changes on Wikitext2.

| Model | Method | ARC-c | ARC-e | BoolQ | HellaS | Lam. | OBQA | PIQA | SIQA | WinoG. | Avg. | Wiki2 PPL |
|---|---|---|---|---|---|---|---|---|---|---|---|---|
| LLaMA3-8B | RTN | 23.72 | 30.56 | 46.18 | 29.83 | 2.70 | 28.60 | 52.45 | 34.39 | 50.20 | 33.18 | 704.34 |
| | SpinQuant | 46.33 | 73.57 | 76.15 | 75.43 | 71.40 | 41.40 | 79.16 | 44.68 | 68.75 | 64.10 | 7.35 |
| | SpinQuant + KL-Top | 47.29 | 73.95 | 75.82 | 75.64 | 71.40 | 41.58 | 78.16 | 44.38 | 68.45 | 64.07 | 7.54 |
| | OSTQuant | 49.26 | 76.68 | 78.25 | 76.18 | 70.48 | 43.19 | 77.85 | 45.18 | 69.13 | 65.13 | 6.80 |
| | OSTQuant + KL-Top | 49.32 | 76.73 | 78.87 | 76.01 | 70.77 | 43.20 | 78.51 | 45.70 | 69.22 | 65.37 | 7.29 |
| LLaMA2-7B | RTN | 27.22 | 27.06 | 50.83 | 27.34 | 0.93 | 25.80 | 49.51 | 34.85 | 50.51 | 32.67 | nan |
| | SpinQuant | 40.44 | 71.08 | 74.40 | 73.51 | 70.66 | 41.80 | 76.88 | 43.50 | 65.82 | 62.01 | 5.96 |
| | SpinQuant + KL-Top | 40.76 | 71.29 | 74.61 | 73.08 | 70.19 | 40.94 | 76.32 | 43.85 | 67.78 | 62.09 | 6.16 |
| | OSTQuant | 42.41 | 69.87 | 75.07 | 72.90 | 70.21 | 40.87 | 78.16 | 44.16 | 68.40 | 62.45 | 5.38 |
| | OSTQuant + KL-Top | 44.62 | 72.69 | 75.41 | 73.27 | 70.21 | 41.00 | 78.13 | 44.42 | 68.27 | 63.11 | 5.94 |

Table 9: Prefill time and Memory usage of LLaMA models with different parameter sizes and sequence lengths, compared between our 4-bit implementation and FP16. All tests were conducted on a Transformer block with batch size 4 on a 3090 GPU.

| Model | Seqlen | Prefill Time | | Prefill Speedup | Memory | | Memory Saving |
|---|---|---|---|---|---|---|---|
| | | FP16 | INT4 | | FP16 | INT4 | |
| LLaMA2-7B | 256 | 8.050ms | 3.597ms | 2.238x | 0.411GB | 0.118GB | 3.479x |
| | 512 | 14.904ms | 6.579ms | 2.265x | 0.435GB | 0.130GB | 3.341x |
| | 1024 | 27.989ms | 12.582ms | 2.225x | 0.483GB | 0.155GB | 3.116x |
| | 2048 | 54.276ms | 25.312ms | 2.144x | 0.577GB | 0.202GB | 2.857x |
| | 4096 | 112.230ms | 53.145ms | 2.112x | 0.766GB | 0.299GB | 2.566x |
| | 8192 | 244.675ms | 121.339ms | 2.016x | 1.147GB | 0.491GB | 2.336x |
| LLaMA3-8B | 256 | 8.035ms | 3.314ms | 2.424x | 0.430GB | 0.124GB | 3.478x |
| | 512 | 15.545ms | 6.176ms | 2.517x | 0.442GB | 0.132GB | 3.356x |
| | 1024 | 29.169ms | 11.599ms | 2.515x | 0.466GB | 0.149GB | 3.116x |
| | 2048 | 57.470ms | 23.631ms | 2.432x | 0.513GB | 0.185GB | 2.774x |
| | 4096 | 117.523ms | 49.835ms | 2.358x | 0.608GB | 0.256GB | 2.378x |
| | 8192 | 256.394ms | 114.815ms | 2.233x | 0.795GB | 0.397GB | 2.003x |
| LLaMA2-13B | 256 | 11.449ms | 4.370ms | 2.620x | 0.634GB | 0.174GB | 3.643x |
| | 512 | 21.195ms | 7.924ms | 2.675x | 0.663GB | 0.189GB | 3.512x |
| | 1024 | 41.752ms | 15.867 ms | 2.631x | 0.723GB | 0.219GB | 3.301x |
| | 2048 | 81.965ms | 32.553ms | 2.518x | 0.841GB | 0.279GB | 3.018x |
| | 4096 | 199.046ms | 70.442ms | 2.826x | 1.079GB | 0.399GB | 2.702x |
| | 8192 | 359.409ms | 154.640ms | 2.324x | 1.551GB | 0.639GB | 2.426x |
| LLaMA-30B | 256 | 18.682ms | 5.883ms | 3.175x | 1.047GB | 0.283GB | 3.703x |
| | 512 | 34.393ms | 11.445ms | 3.005x | 1.085GB | 0.302GB | 3.589x |
| | 1024 | 66.880ms | 22.464ms | 2.977x | 1.162GB | 0.340GB | 3.416x |
| | 2048 | 157.500ms | 46.317ms | 3.400x | 1.315GB | 0.418GB | 3.148x |
| | 4096 | 272.355ms | 96.052ms | 2.835x | 1.625GB | 0.575GB | 2.828x |
| | 8192 | 576.555ms | 215.27ms | 2.678x | 2.242GB | 0.887GB | 2.527x |

we can see, quantizing the weights to below 4 bits leads to a significant drop in accuracy. Another notable observation is that larger models, such as LLaMA-3-70B, outperform smaller floating-point models like LLaMA-30B in both accuracy and parameter size after quantization.

As shown in Tab 10, since the decoding stage of LLMs is often memory-bound, quantizing the weights to 4-bit significantly reduces memory access overhead, thereby accelerating the inference process. The results demonstrate that, under the premise of achieving nearly lossless accuracy, even a single A6000 GPU can successfully run LLaMA-3-70B at a speed of 15 tokens per second after quantization. This validates the effectiveness of our quantization method.

Tab 11 presents a comparison of training time between our approach and block reconstruction-based methods across models of varying parameter sizes. Thanks to the effectiveness of Weight Outlier Minimization Initialization and KL-Top Loss, we require only about 150 iterations to complete the quantization process. As shown in the table, OSTQuant offers a significant optimization speed advantage for smaller models, such as achieving more than a 5x speedup on the LLaMA 7B model compared to OmniQuant. Even for large-scale models like LLaMA 70B, it achieves nearly a 2x speed improvement.

Table 10: Comparison of generation speed and memory usage before and after quantization in the decoding stage.

| Model | Decoder Speed (tokens/sec) | | | Memory Use (GB) | | Memory Saving |
|---|---|---|---|---|---|---|
| | FP | Quantized | Speed up | FP | Quantized | |
| LLaMA-2-7B | 47.32 | 89.4 | 1.89x | 13.94 | 4.32 | 3.23x |
| LLaMA-3-8B | 38.33 | 77.71 | 2.03x | 15.83 | 5.88 | 2.69x |
| LLaMA-2-13B | 23.7 | 55.35 | 2.34x | 23.7 | 8.5 | 2.79x |
| LLaMA-30B | OOM | 30.49 | - | OOM | 18.19 | - |
| LLaMA-3-70B | OOM | 14.68 | - | OOM | 38.41 | - |

Table 11: Comparison of the training time between our approach and block reconstruction-based methods across models with varying parameter sizes.

| Method | 7B | 8B | 13B | 30B | 70B |
|---|---|---|---|---|---|
| Omniquant | 1.6h | 1.8h | 3.3h | 7.3h | 9.5h |
| OSTQuant | 0.3h | 0.4h | 0.8h | 2.2h | 5.5h |
| Speedup | 5.3x | 4.5x | 4.1x | 3.3x | 1.7x |

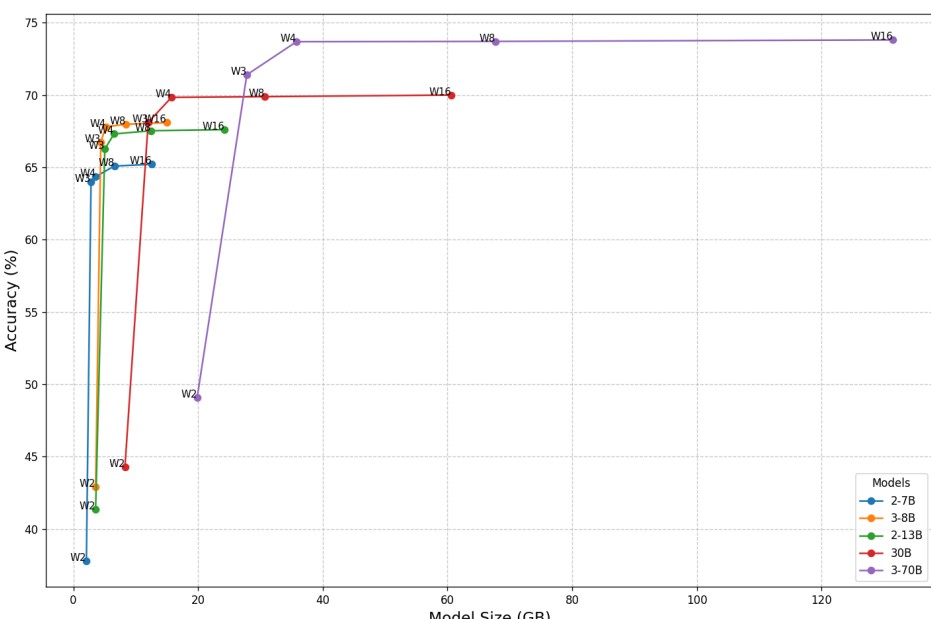

Figure 8: Performance and model size of the Llama series under different quantization settings. Accuracy is calculated by the average score of nine zero-shot datasets. Model size is determined by the memory usage when stored in fp16 or integer formats.

## A.5 FUTURE WORK

### A.5.1 OSTQUANT FOR FULLY-QUANT LARGE LANGUAGE MODELS

Fig 9 introduces OSTQuant's novel strategy designed for full quantization. Full quantization involves quantizing all activations within each Transformer Block to low bits (as shown by the quantization nodes inserted for all node inputs and outputs in the figure). This reduces memory transfer overhead for activations and fully utilizes efficient low-precision computational units.

As shown in Fig 9, OSTQuant introduces numerous equivalence transformations to alter the distributions of all node input and output activations. Unlike the methods in Fig 5 designed for traditional quantization, the design for fully quantization adds more equivalence transformations, particularly around ROPE and SiLU. Specifically:

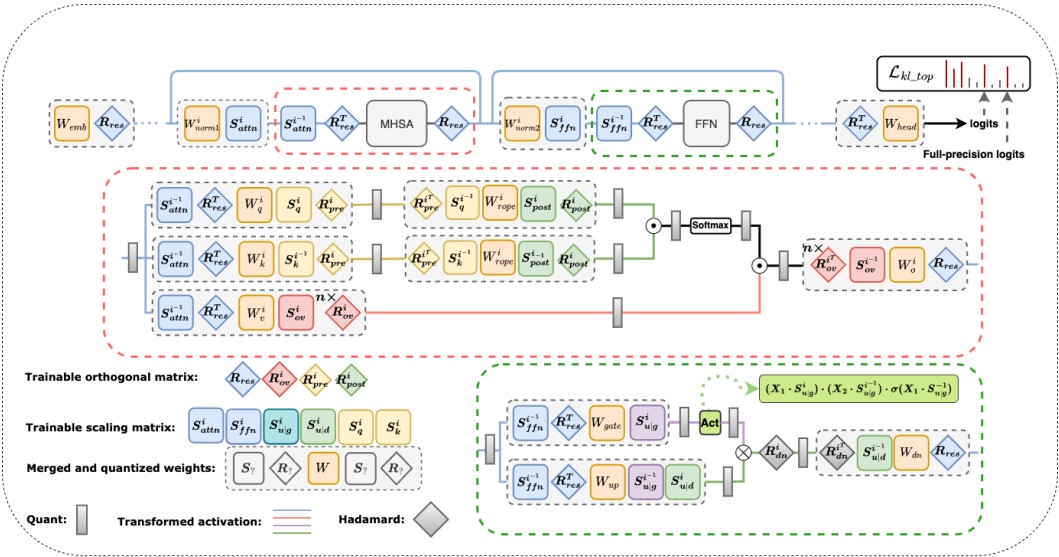

Figure 9: The extension of OSTQuant for full quantization.

1. **ROPE Handling**: We treat ROPE as a lightweight GEMM layer and construct a weight matrix of shape $(\text{token}, \text{head\_dim}, \text{head\_dim})$ based on its principle. We then introduce pre-ROPE and post-ROPE transformation pairs.

   a. Pre-ROPE transformations are based on the fact that ROPE and the preceding linear layer can be viewed as consecutive matrix multiplications along the head_dim. The corresponding transformation pairs are represented in figure by $S_q^i R_{pre}^i$ nd $S_k^i R_{pre}^i$.

   b. Post-ROPE transformations rely on the attention computation formula $\boldsymbol{Attn} = \boldsymbol{Q} @ \boldsymbol{K}^\top$, where $\boldsymbol{Q}$ means the query matrix and $\boldsymbol{K}$ means the key matrix in Self-Attention module. The corresponding transformation pairs are represented in figure by $S_{post}^i R_{post}^i$

2. **Smoothing Activation Discrepancies of SiLU**: Inspired by smoothing methods for SwiGLU in I-LLM (Hu et al., 2024), we decompose SiLU as $\text{SiLU}(\boldsymbol{X}) = \boldsymbol{X} \cdot \sigma(\boldsymbol{X})$ and use equivalences such as

$$\boldsymbol{X_1} \cdot \boldsymbol{X_2} \cdot \sigma(\boldsymbol{X_1}) = (\boldsymbol{X_1} \cdot \boldsymbol{S}) \cdot (\boldsymbol{S_2} \cdot \frac{1}{\boldsymbol{S}}) \cdot \sigma((\boldsymbol{X_1} \cdot \boldsymbol{S}) \cdot \frac{1}{\boldsymbol{S}})$$

to alleviate inter-channel discrepancies of activations before and after SiLU. The corresponding transformation is represented in figure by $\boldsymbol{S_{u|g}}$.

We will conduct experiments in full-quantization domain in the future to fully explore the potential of OSTQuant.

## A.6 FULL RESULTS

### A.6.1 QUANTITATIVE RESULTS

In this section, we provide a comprehensive presentation of our results across various datasets to complement the main paper. Specifically, the results include:

- Complete comparison of the perplexity score on WikiText2 and averaged accuracy on zero-shot common sense reasoning tasks on LLaMA-1(Tab 13), 2 and 3 (Tab 12).

- Validate the effectiveness of OSTQuant on larger-scale language models such as LLaMA-2-70B and LLaMA-3-70B(Tab 14).

Table 12: Complete omparison of the perplexity score on WikiText2 and averaged accuracy on Zero-shot Common Sense Reasoning tasks on **LLaMA-2 & 3**.

| Model | #Bits W-A-KV | Method | ARC-c (↑) | ARC-e (↑) | BoolQ (↑) | HellaS. (↑) | Lam. (↑) | OBQA (↑) | PIQA (↑) | SIQA (↑) | WinoG. (↑) | Avg. (↑) | Wiki2 (↓) |
|---|---|---|---|---|---|---|---|---|---|---|---|---|---|
| 2-7B | 16-16-16 | Full Precision | 46.42 | 74.33 | 77.71 | 75.94 | 73.69 | 44.20 | 79.16 | 45.91 | 69.53 | 65.21 | 5.47 |
| | 4-16-16 | RTN | 42.15 | 67.59 | 73.06 | 72.34 | 67.18 | 41.80 | 76.50 | 44.11 | 66.69 | 61.27 | 7.02 |
| | | SmoothQuant | 39.59 | 65.19 | 69.82 | 68.84 | 62.27 | 40.20 | 75.95 | 44.17 | 63.85 | 58.88 | 8.03 |
| | | GPTQ | 42.49 | 69.53 | 61.31 | 73.83 | 67.61 | 42.40 | 77.64 | 44.52 | 68.43 | 60.86 | 9.84 |
| | | Omniquant | 42.49 | 71.00 | 74.34 | 73.85 | 70.70 | 44.20 | 78.40 | 44.93 | 68.82 | 63.19 | 5.74 |
| | | AWQ | 44.11 | 70.75 | 78.07 | 74.98 | 70.68 | 43.80 | 78.13 | 45.14 | 69.38 | 63.89 | 5.83 |
| | | QuaRot | 43.94 | 73.15 | 76.97 | 74.87 | 73.06 | 44.00 | 78.24 | 45.09 | 69.38 | 64.30 | 5.62 |
| | | SpinQuant | 43.34 | 72.69 | 73.36 | 75.10 | 73.80 | 43.00 | 77.86 | 45.60 | 67.56 | 63.59 | 5.58 |
| | | **OSTQuant** | 44.54 | 73.31 | 75.57 | 75.04 | 73.67 | 44.20 | 78.89 | 45.50 | 68.59 | 64.37 | 5.64 |
| | 4-4-16 | RTN | 25.34 | 28.03 | 50.52 | 27.71 | 1.01 | 26.20 | 50.82 | 33.93 | 48.38 | 32.44 | nan |
| | | SmoothQuant | 28.33 | 26.39 | 49.39 | 27.28 | 1.18 | 23.40 | 48.80 | 33.62 | 50.75 | 32.13 | nan |
| | | GPTQ | 24.40 | 28.70 | 51.62 | 28.66 | 1.36 | 24.60 | 51.14 | 34.49 | 49.49 | 32.72 | nan |
| | | QuaRot | 42.32 | 69.65 | 74.77 | 72.91 | 70.81 | 39.80 | 77.20 | 43.55 | 65.82 | 61.87 | 6.05 |
| | | SpinQuant | 37.54 | 62.58 | 71.16 | 70.48 | 67.16 | 34.80 | 75.46 | 39.76 | 60.62 | 57.37 | 6.78 |
| | | **OSTQuant** | 44.03 | 71.93 | 75.41 | 74.94 | 73.22 | 43.20 | 78.51 | 45.85 | 68.03 | 63.90 | 5.60 |
| | 4-4-4 | RTN | 27.22 | 27.06 | 50.83 | 27.34 | 0.93 | 25.80 | 49.51 | 34.85 | 50.51 | 32.67 | nan |
| | | SmoothQuant | 26.37 | 25.63 | 47.71 | 27.05 | 1.11 | 26.40 | 51.90 | 34.49 | 48.38 | 32.12 | nan |
| | | GPTQ | 26.96 | 27.65 | 52.84 | 28.83 | 1.63 | 29.20 | 49.62 | 35.11 | 49.80 | 33.52 | nan |
| | | Omniquant | 31.40 | 53.75 | 63.79 | 55.06 | 35.63 | 34.40 | 66.59 | 40.28 | 54.70 | 48.40 | 14.26 |
| | | QuaRot | 41.43 | 69.32 | 74.19 | 72.50 | 70.66 | 39.80 | 77.42 | 43.35 | 64.64 | 61.48 | 6.11 |
| | | SpinQuant | 40.44 | 71.08 | 74.40 | 73.51 | 70.66 | 41.80 | 76.88 | 43.50 | 65.82 | 62.01 | 5.96 |
| | | **OSTQuant** | 42.92 | 72.56 | 74.71 | 73.14 | 71.76 | 44.40 | 77.42 | 44.98 | 66.77 | 63.18 | 5.91 |
| 2-13B | 16-16-16 | Full Precision | 49.15 | 77.53 | 80.58 | 79.39 | 76.62 | 45.20 | 80.63 | 47.49 | 71.90 | 67.61 | 4.88 |
| | 4-16-16 | RTN | 42.92 | 66.54 | 71.38 | 66.62 | 68.99 | 39.40 | 76.93 | 44.06 | 65.35 | 60.24 | 6.39 |
| | | SmoothQuant | 46.25 | 70.45 | 74.92 | 69.16 | 70.49 | 39.80 | 77.86 | 45.14 | 64.17 | 62.03 | 5.86 |
| | | GPTQ | 49.63 | 73.95 | 74.83 | 73.77 | 73.20 | 42.40 | 78.51 | 45.50 | 70.64 | 64.71 | 5.79 |
| | | Omniquant | 48.29 | 75.42 | 77.92 | 77.80 | 75.59 | 45.20 | 80.41 | 46.62 | 70.17 | 66.38 | 5.02 |
| | | AWQ | 48.63 | 78.16 | 78.81 | 78.48 | 75.20 | 45.00 | 79.54 | 46.21 | 72.45 | 66.25 | 5.07 |
| | | QuaRot | 49.15 | 76.26 | 80.46 | 78.17 | 76.50 | 45.40 | 80.03 | 45.50 | 71.11 | 66.95 | 5.00 |
| | | SpinQuant | 49.15 | 77.48 | 79.27 | 78.46 | 77.10 | 44.60 | 80.03 | 46.47 | 71.67 | 67.14 | 5.00 |
| | | **OSTQuant** | 48.72 | 76.26 | 80.67 | 78.27 | 76.54 | 45.54 | 80.25 | 47.65 | 71.90 | 67.31 | 4.94 |
| | 4-4-16 | RTN | 27.99 | 26.81 | 38.50 | 26.08 | 0.00 | 23.60 | 48.20 | 34.90 | 51.62 | 30.86 | 8e3 |
| | | SmoothQuant | 24.49 | 35.06 | 47.98 | 30.87 | 3.67 | 26.20 | 55.01 | 35.31 | 49.72 | 34.26 | 1e3 |
| | | GPTQ | 27.82 | 26.77 | 37.92 | 25.67 | 0.00 | 21.80 | 47.77 | 35.11 | 48.15 | 30.11 | 4e3 |
| | | QuaRot | 46.42 | 73.86 | 78.10 | 75.68 | 74.31 | 43.00 | 79.05 | 44.37 | 71.35 | 65.13 | 5.35 |
| | | SpinQuant | 43.77 | 69.99 | 76.57 | 74.63 | 72.81 | 41.60 | 77.20 | 44.27 | 68.19 | 63.23 | 5.24 |
| | | **OSTQuant** | 47.78 | 74.66 | 80.03 | 77.60 | 75.94 | 44.40 | 79.38 | 46.06 | 70.32 | 66.24 | 5.14 |
| | 4-4-4 | RTN | 27.82 | 26.52 | 38.38 | 26.27 | 0.02 | 26.00 | 49.78 | 34.39 | 49.17 | 30.93 | 7e3 |
| | | SmoothQuant | 24.49 | 33.00 | 45.84 | 30.70 | 2.70 | 23.80 | 53.81 | 34.80 | 51.07 | 33.36 | 2e3 |
| | | GPTQ | 27.90 | 26.39 | 37.95 | 26.16 | 0.00 | 27.00 | 48.26 | 34.39 | 50.43 | 27.85 | 5e3 |
| | | Omniquant | 32.85 | 55.13 | 64.34 | 60.13 | 42.85 | 33.40 | 68.17 | 39.76 | 56.51 | 50.35 | 12.30 |
| | | QuaRot | 47.27 | 73.91 | 78.41 | 75.33 | 73.53 | 43.80 | 79.27 | 45.85 | 69.06 | 65.16 | 5.39 |
| | | SpinQuant | 46.67 | 74.49 | 76.76 | 75.22 | 72.19 | 42.40 | 78.29 | 43.45 | 67.72 | 64.13 | 5.74 |
| | | **OSTQuant** | 47.10 | 75.21 | 77.46 | 76.71 | 75.14 | 44.60 | 78.67 | 45.75 | 68.03 | 65.41 | 5.25 |
| 3-8B | 16-16-16 | Full Precision | 53.50 | 77.74 | 81.10 | 79.18 | 75.74 | 44.80 | 80.63 | 47.08 | 73.01 | 68.09 | 6.14 |
| | 4-16-16 | RTN | 48.98 | 73.23 | 72.75 | 75.90 | 63.85 | 43.20 | 78.40 | 43.81 | 73.16 | 63.70 | 8.13 |
| | | SmoothQuant | 47.44 | 72.35 | 72.11 | 74.92 | 62.41 | 43.00 | 77.69 | 43.91 | 71.27 | 62.79 | 8.12 |
| | | GPTQ | 49.74 | 72.52 | 71.28 | 68.34 | 46.69 | 43.60 | 78.78 | 46.47 | 71.82 | 61.03 | 7.43 |
| | | Omniquant | 50.09 | 74.54 | 79.51 | 76.92 | 70.31 | 43.80 | 79.54 | 44.52 | 71.74 | 65.66 | 7.19 |
| | | AWQ | 52.22 | 76.68 | 80.51 | 77.51 | 74.81 | 44.20 | 79.60 | 46.26 | 71.67 | 67.03 | 7.36 |
| | | QuaRot | 51.88 | 77.53 | 79.60 | 77.87 | 74.11 | 44.40 | 80.14 | 46.37 | 73.56 | 67.27 | 6.53 |
| | | SpinQuant | 52.13 | 72.28 | 79.20 | 78.40 | 73.76 | 44.80 | 79.98 | 45.50 | 72.77 | 66.54 | 6.49 |
| | | **OSTQuant** | 52.72 | 79.84 | 80.31 | 77.86 | 76.48 | 42.80 | 80.74 | 45.55 | 73.80 | 67.80 | 6.53 |
| | 4-4-16 | RTN | 23.72 | 30.89 | 46.30 | 31.26 | 3.03 | 27.60 | 52.72 | 35.26 | 50.04 | 33.42 | 6e2 |
| | | SmoothQuant | 23.29 | 28.28 | 48.93 | 29.19 | 1.57 | 28.60 | 54.46 | 33.37 | 49.64 | 33.04 | 1e3 |
| | | GPTQ | 23.46 | 32.07 | 43.79 | 30.10 | 2.41 | 28.00 | 53.97 | 34.14 | 48.86 | 32.98 | 6e2 |
| | | QuaRot | 42.66 | 67.26 | 73.73 | 73.60 | 67.42 | 43.00 | 76.61 | 45.04 | 65.90 | 61.69 | 8.02 |
| | | SpinQuant | 47.35 | 74.12 | 76.36 | 75.98 | 69.88 | 42.46 | 77.37 | 44.47 | 68.98 | 64.11 | 7.28 |
| | | **OSTQuant** | 48.81 | 73.48 | 79.82 | 75.97 | 72.62 | 42.40 | 78.18 | 45.75 | 69.22 | 65.14 | 7.24 |
| | 4-4-4 | RTN | 23.72 | 30.56 | 46.18 | 29.83 | 2.70 | 28.60 | 52.45 | 34.39 | 50.20 | 33.18 | 7e2 |
| | | SmoothQuant | 23.55 | 28.96 | 48.84 | 28.90 | 1.44 | 29.40 | 51.09 | 34.14 | 50.36 | 32.96 | 1e3 |
| | | GPTQ | 23.38 | 32.74 | 44.34 | 29.72 | 2.39 | 29.80 | 54.95 | 34.75 | 51.30 | 33.71 | 6e2 |
| | | Omniquant | 22.87 | 30.35 | 41.53 | 31.11 | 1.86 | 25.40 | 53.37 | 34.08 | 50.43 | 32.33 | 4e2 |
| | | QuaRot | 42.83 | 67.42 | 73.21 | 72.66 | 66.93 | 42.20 | 75.73 | 45.19 | 66.22 | 61.38 | 8.18 |
| | | SpinQuant | 46.33 | 73.57 | 76.15 | 75.43 | 71.40 | 41.40 | 79.16 | 44.68 | 68.75 | 64.10 | 7.35 |
| | | **OSTQuant** | 49.32 | 76.73 | 78.87 | 76.01 | 70.77 | 43.20 | 78.51 | 45.70 | 69.22 | 65.37 | 7.29 |

Table 13: Complete omparison of the perplexity score on WikiText2 and averaged accuracy on Zero-shot Common Sense Reasoning tasks on **LLaMA**.

| Model | #Bits W-A-KV | Method | ARC-c (↑) | ARC-e (↑) | BoolQ (↑) | HellaS. (↑) | LambA. (↑) | OBQA (↑) | PIQA (↑) | SIQA (↑) | WinoG. (↑) | Avg. (↑) | Wiki2 (↓) |
|---|---|---|---|---|---|---|---|---|---|---|---|---|---|
| 7B | 16-16-16 | Full Precision | 44.71 | 72.90 | 74.98 | 76.20 | 73.08 | 43.80 | 79.16 | 45.55 | 69.93 | 64.48 | 5.68 |
| | 4-16-16 | RTN | 43.17 | 69.82 | 73.30 | 73.75 | 69.67 | 42.00 | 78.13 | 45.34 | 68.82 | 62.67 | 7.94 |
| | | SmoothQuant | 40.96 | 68.60 | 74.04 | 73.16 | 68.74 | 42.00 | 78.07 | 46.11 | 68.51 | 62.24 | 7.46 |
| | | GPTQ | 41.72 | 67.85 | 67.98 | 69.50 | 63.15 | 40.80 | 76.55 | 44.37 | 69.46 | 60.15 | 7.93 |
| | | Omniquant | 42.49 | 71.38 | 74.62 | 74.71 | 71.98 | 42.00 | 79.05 | 45.96 | 68.59 | 63.42 | 5.86 |
| | | AWQ | 43.86 | 70.79 | 74.19 | 75.27 | 69.94 | 43.00 | 78.45 | 45.09 | 69.14 | 63.30 | 5.97 |
| | | QuaRot | 42.75 | 69.99 | 73.30 | 75.13 | 73.55 | 42.80 | 78.35 | 45.14 | 69.61 | 63.40 | 5.83 |
| | | SpinQuant | 43.77 | 71.17 | 74.46 | 75.09 | 72.91 | 44.40 | 78.40 | 44.52 | 70.72 | 63.94 | 5.76 |
| | | **OSTQuant** | 44.20 | 72.56 | 73.73 | 75.05 | 73.45 | 44.60 | 78.73 | 45.45 | 69.38 | 64.13 | 5.81 |
| | 4-4-16 | RTN | 23.46 | 29.34 | 45.05 | 29.02 | 1.24 | 26.00 | 52.07 | 35.11 | 51.30 | 32.51 | 7e3 |
| | | SmoothQuant | 25.17 | 31.40 | 51.62 | 29.73 | 5.43 | 28.20 | 54.68 | 34.44 | 49.09 | 34.42 | 3e2 |
| | | GPTQ | 23.89 | 27.74 | 42.87 | 28.49 | 1.28 | 27.40 | 51.00 | 36.23 | 50.20 | 32.12 | 1e3 |
| | | QuaRot | 40.36 | 67.26 | 73.15 | 72.89 | 70.81 | 42.00 | 77.97 | 44.27 | 67.17 | 61.76 | 6.22 |
| | | SpinQuant | 40.19 | 68.43 | 72.35 | 72.91 | 70.68 | 41.20 | 77.75 | 44.17 | 68.67 | 61.82 | 6.08 |
| | | **OSTQuant** | 42.58 | 70.79 | 72.87 | 74.06 | 70.77 | 43.40 | 77.69 | 45.04 | 67.25 | 62.72 | 6.04 |
| | 4-4-4 | RTN | 23.89 | 29.59 | 46.67 | 28.37 | 1.13 | 26.40 | 52.99 | 35.21 | 51.54 | 32.87 | 1e4 |
| | | SmoothQuant | 23.38 | 30.18 | 50.03 | 29.67 | 4.89 | 24.60 | 51.74 | 34.75 | 50.67 | 33.32 | 3e2 |
| | | GPTQ | 23.89 | 27.90 | 43.88 | 27.86 | 1.05 | 26.20 | 51.85 | 34.08 | 49.49 | 31.80 | 2e3 |
| | | Omniquant | 31.40 | 54.84 | 61.80 | 56.98 | 38.29 | 31.80 | 66.59 | 39.30 | 55.17 | 48.46 | 11.26 |
| | | QuaRot | 40.27 | 67.55 | 72.20 | 72.59 | 70.62 | 39.80 | 77.20 | 44.88 | 65.90 | 61.22 | 6.26 |
| | | SpinQuant | 39.08 | 68.18 | 73.06 | 72.87 | 70.46 | 40.60 | 77.42 | 42.68 | 67.56 | 61.32 | 6.12 |
| | | **OSTQuant** | 42.92 | 70.33 | 72.11 | 73.77 | 70.66 | 42.42 | 77.91 | 44.93 | 67.88 | 62.55 | 6.07 |
| 13B | 16-16-16 | Full Precision | 47.87 | 74.49 | 77.86 | 79.10 | 76.03 | 44.40 | 80.30 | 46.72 | 73.24 | 66.67 | 5.09 |
| | 4-16-16 | RTN | 45.56 | 70.66 | 72.45 | 76.06 | 70.58 | 42.00 | 78.84 | 44.93 | 70.01 | 63.45 | 8.60 |
| | | SmoothQuant | 43.86 | 71.21 | 71.62 | 74.19 | 69.34 | 40.00 | 77.80 | 45.45 | 70.72 | 62.69 | 18.75 |
| | | GPTQ | 45.99 | 72.85 | 73.27 | 75.31 | 70.10 | 44.60 | 79.87 | 46.16 | 71.11 | 64.36 | 6.58 |
| | | Omniquant | 47.01 | 73.86 | 77.22 | 77.95 | 75.59 | 45.00 | 79.87 | 46.88 | 72.61 | 66.22 | 5.21 |
| | | AWQ | 47.53 | 73.86 | 75.60 | 59.03 | 78.34 | 43.40 | 79.87 | 45.85 | 71.67 | 65.58 | 5.28 |
| | | QuaRot | 47.18 | 72.22 | 76.85 | 78.07 | 75.99 | 45.00 | 79.76 | 45.70 | 72.38 | 65.91 | 5.20 |
| | | SpinQuant | 47.44 | 74.83 | 77.37 | 78.13 | 75.55 | 45.60 | 79.92 | 46.01 | 72.06 | 66.32 | 5.16 |
| | | **OSTQuant** | 48.04 | 73.86 | 78.10 | 78.28 | 75.99 | 45.60 | 80.52 | 46.93 | 72.30 | 66.62 | 5.21 |
| | 4-4-16 | RTN | 25.85 | 26.26 | 42.05 | 26.70 | 0.17 | 28.00 | 50.33 | 34.60 | 50.67 | 31.63 | 2e4 |
| | | SmoothQuant | 25.43 | 29.29 | 51.56 | 28.12 | 2.02 | 26.00 | 53.32 | 34.34 | 49.57 | 33.29 | 6e2 |
| | | GPTQ | 24.66 | 27.78 | 40.80 | 25.83 | 0.70 | 24.20 | 51.31 | 36.65 | 50.20 | 31.51 | 3e3 |
| | | QuaRot | 46.93 | 71.51 | 75.57 | 76.63 | 74.13 | 42.40 | 78.73 | 45.24 | 68.98 | 64.46 | 5.50 |
| | | SpinQuant | 45.73 | 72.56 | 75.38 | 76.86 | 73.28 | 43.60 | 78.89 | 44.63 | 70.40 | 64.59 | 5.36 |
| | | **OSTQuant** | 48.04 | 74.07 | 77.13 | 77.22 | 74.58 | 45.00 | 78.62 | 46.16 | 71.35 | 65.80 | 5.40 |
| | 4-4-4 | RTN | 26.28 | 27.27 | 42.35 | 25.85 | 0.19 | 26.60 | 49.95 | 34.19 | 50.67 | 31.33 | 3e4 |
| | | SmoothQuant | 24.49 | 28.83 | 51.65 | 27.91 | 2.08 | 26.00 | 52.56 | 35.41 | 50.59 | 33.28 | 5e2 |
| | | GPTQ | 23.63 | 27.31 | 39.85 | 26.17 | 0.56 | 26.00 | 51.96 | 35,82 | 49.57 | 30.63 | 3e3 |
| | | Omniquant | 29.61 | 48.23 | 58.20 | 56.45 | 28.76 | 31.40 | 65.29 | 37.10 | 55.64 | 45.63 | 10.87 |
| | | QuaRot | 46.50 | 71.55 | 75.08 | 76.43 | 73.47 | 45.00 | 78.78 | 44.37 | 70.09 | 64.59 | 5.53 |
| | | SpinQuant | 45.99 | 70.71 | 76.51 | 77.16 | 73.63 | 45.60 | 79.00 | 45.65 | 70.32 | 64.95 | 5.39 |
| | | **OSTQuant** | 45.90 | 75.25 | 76.94 | 77.21 | 74.23 | 43.40 | 79.43 | 45.91 | 70.56 | 65.43 | 5.40 |
| 30B | 16-16-16 | Full Precision | 52.99 | 80.39 | 82.75 | 82.62 | 77.59 | 48.00 | 82.26 | 47.75 | 75.69 | 70.00 | 4.10 |
| | 4-16-16 | RTN | 49.74 | 73.99 | 77.89 | 79.07 | 72.21 | 44.20 | 79.00 | 45.70 | 73.88 | 65.69 | 6.13 |
| | | SmoothQuant | 48.98 | 72.94 | 80.00 | 79.00 | 71.49 | 44.80 | 78.13 | 45.96 | 73.16 | 65.69 | 5.80 |
| | | GPTQ | 50.85 | 75.97 | 80.31 | 79.31 | 74.13 | 45.00 | 78.94 | 45.24 | 72.77 | 66.95 | 5.26 |
| | | Omniquant | 52.22 | 78.62 | 81.80 | 81.94 | 76.85 | 47.20 | 81.07 | 47.54 | 74.43 | 69.07 | 4.25 |
| | | AWQ | 53.24 | 77.48 | 81.68 | 82.29 | 76.79 | 48.20 | 81.72 | 48.16 | 75.37 | 69.44 | 4.28 |
| | | QuaRot | 53.58 | 78.62 | 82.11 | 82.10 | 77.62 | 48.00 | 81.72 | 47.75 | 76.09 | 69.73 | 4.27 |
| | | SpinQuant | 52.90 | 78.49 | 82.02 | 82.21 | 78.28 | 48.20 | 81.01 | 48.41 | 75.06 | 69.62 | 4.21 |
| | | **OSTQuant** | 53.07 | 79.12 | 83.09 | 82.04 | 78.58 | 48.60 | 81.18 | 48.06 | 74.82 | 69.84 | 4.19 |
| | 4-4-16 | RTN | 25.00 | 27.95 | 42.02 | 27.22 | 0.21 | 27.00 | 49.13 | 34.65 | 50.91 | 31.57 | 2e3 |
| | | SmoothQuant | 23.63 | 30.68 | 54.86 | 31.91 | 3.80 | 28.20 | 54.13 | 34.49 | 50.04 | 34.64 | 1e3 |
| | | GPTQ | 27.30 | 27.19 | 38.69 | 26.75 | 0.17 | 25.80 | 49.02 | 35.21 | 47.75 | 30.88 | 2e3 |
| | | QuaRot | 51.79 | 76.39 | 80.76 | 80.90 | 77.08 | 45.80 | 80.58 | 45.60 | 74.35 | 68.14 | 4.57 |
| | | SpinQuant | 50.06 | 77.06 | 81.38 | 80.62 | 76.79 | 46.00 | 80.14 | 46.37 | 74.27 | 68.08 | 4.53 |
| | | **OSTQuant** | 51.37 | 78.11 | 82.48 | 79.51 | 75.99 | 45.40 | 81.18 | 47.80 | 74.82 | 68.52 | 4.43 |
| | 4-4-4 | RTN | 25.00 | 28.87 | 44.07 | 27.29 | 0.39 | 25.60 | 49.67 | 34.54 | 49.33 | 31.64 | 2e3 |
| | | SmoothQuant | 22.61 | 32.87 | 55.05 | 31.28 | 3.40 | 28.00 | 53.75 | 34.65 | 50.28 | 34.65 | 1e3 |
| | | GPTQ | 27.22 | 27.82 | 39.36 | 27.13 | 0.33 | 24.80 | 50.71 | 34.34 | 47.91 | 31.07 | 2e3 |
| | | Omniquant | 29.10 | 53.79 | 54.95 | 52.44 | 26.45 | 30.60 | 65.56 | 38.54 | 53.91 | 45.04 | 10.33 |
| | | QuaRot | 51.71 | 76.98 | 80.95 | 80.86 | 77.04 | 46.20 | 80.63 | 45.00 | 73.32 | 68.08 | 4.60 |
| | | SpinQuant | 51.62 | 76.98 | 81.07 | 80.57 | 76.63 | 46.00 | 79.92 | 46.26 | 74.19 | 68.14 | 4.55 |
| | | **OSTQuant** | 49.74 | 76.52 | 81.16 | 81.13 | 77.57 | 46.40 | 80.90 | 46.11 | 74.27 | 68.20 | 4.42 |

Table 14: Complete comparison of the perplexity score on WikiText2 and averaged accuracy on Zero-shot Common Sense Reasoning tasks for **LLaMA2-70B and LLaMA3-70B**.

| Model | #Bits W-A-KV | Method | ARC-c (↑) | ARC-e (↑) | BoolQ (↑) | HellaS. (↑) | LambA. (↑) | OBQA (↑) | PIQA (↑) | SIQA (↑) | WinoG. (↑) | Avg. (↑) | Wiki2 (↓) |
|---|---|---|---|---|---|---|---|---|---|---|---|---|---|
| 2-70B | 16-16-16 | Full Precision | 57.42 | 81.02 | 83.79 | 83.81 | 79.60 | 48.80 | 82.70 | 49.18 | 77.98 | 71.59 | 3.32 |
| | 4-16-16 | RTN | 55.80 | 79.29 | 81.35 | 81.78 | 75.51 | 47.60 | 81.94 | 46.83 | 76.48 | 69.62 | 3.87 |
| | | SmoothQuant | 50.26 | 76.56 | 81.53 | 67.81 | 73.63 | 44.40 | 81.34 | 44.17 | 73.64 | 65.93 | 5.50 |
| | | GPTQ | 56.91 | 80.81 | 83.24 | 82.47 | 79.06 | 47.80 | 82.75 | 48.06 | 77.51 | 70.96 | 3.94 |
| | | Omniquant | 57.08 | 80.81 | 82.69 | 83.07 | 79.18 | 47.40 | 83.08 | 48.87 | 77.19 | 71.04 | 3.47 |
| | | AWQ | 56.67 | 80.54 | 82.98 | 82.54 | 78.83 | 47.67 | 82.97 | 48.12 | 77.62 | 70.88 | 4.03 |
| | | QuaRot | 57.34 | 80.85 | 83.24 | 83.27 | 80.38 | 47.60 | 82.21 | 48.62 | 77.35 | 71.21 | 3.41 |
| | | SpinQuant | 56.91 | 80.60 | 83.18 | 83.06 | 79.16 | 49.00 | 82.75 | 48.31 | 77.11 | 71.12 | 3.43 |
| | | **OSTQuant** | 57.36 | 81.37 | 83.20 | 83.86 | 79.77 | 48.73 | 82.69 | 48.46 | 77.89 | 71.48 | 3.41 |
| | 4-4-16 | RTN | 29.35 | 26.05 | 37.74 | 25.97 | 0.02 | 24.80 | 51.31 | 34.14 | 48.70 | 30.90 | 7e4 |
| | | SmoothQuant | 25.00 | 35.98 | 55.23 | 32.52 | 7.49 | 25.00 | 54.62 | 35.21 | 51.70 | 35.86 | 3e2 |
| | | GPTQ | 27.82 | 25.80 | 37.95 | 25.82 | 0.00 | 27.00 | 49.67 | 33.98 | 49.72 | 30.86 | nan |
| | | QuaRot | 55.29 | 80.35 | 81.10 | 81.87 | 79.06 | 45.80 | 82.05 | 47.90 | 76.24 | 69.96 | 3.78 |
| | | SpinQuant | 55.38 | 78.96 | 83.36 | 82.54 | 79.00 | 47.80 | 82.10 | 48.67 | 77.43 | 70.58 | 3.68 |
| | | **OSTQuant** | 56.61 | 80.51 | 83.03 | 82.68 | 79.11 | 47.86 | 83.00 | 48.76 | 76.70 | 70.92 | 3.57 |
| | 4-4-4 | RTN | 30.38 | 27.74 | 38.23 | 26.12 | 0.02 | 24.60 | 51.74 | 34.29 | 52.49 | 31.73 | 7e4 |
| | | SmoothQuant | 24.15 | 33.88 | 55.32 | 31.75 | 7.14 | 26.40 | 54.95 | 34.14 | 52.17 | 35.54 | 3e2 |
| | | GPTQ | 28.75 | 26.39 | 37.86 | 25.96 | 0.00 | 26.40 | 50.00 | 34.44 | 50.04 | 31.09 | nan |
| | | QuaRot | 56.48 | 80.56 | 81.59 | 81.93 | 79.16 | 46.00 | 82.21 | 48.00 | 76.80 | 70.30 | 3.80 |
| | | SpinQuant | 56.31 | 80.64 | 83.55 | 82.36 | 79.41 | 47.20 | 82.21 | 47.29 | 76.16 | 70.57 | 3.61 |
| | | **OSTQuant** | 56.58 | 80.17 | 83.64 | 82.49 | 78.72 | 48.00 | 82.76 | 48.67 | 76.49 | 70.84 | 3.59 |
| 3-70B | 16-16-16 | Full Precision | 64.42 | 85.98 | 85.14 | 84.95 | 79.47 | 48.46 | 84.39 | 50.82 | 80.66 | 73.81 | 2.86 |
| | 4-16-16 | RTN | 26.28 | 25.55 | 37.83 | 26.36 | 0.00 | 29.00 | 50.98 | 34.70 | 49.64 | 31.15 | 1e4 |
| | | SmoothQuant | 51.88 | 77.53 | 80.09 | 80.47 | 73.16 | 46.60 | 80.58 | 45.29 | 75.85 | 67.94 | 6.70 |
| | | GPTQ | 25.77 | 25.29 | 37.83 | 26.36 | 0.12 | 28.40 | 51.74 | 34.90 | 52.64 | 31.45 | 9e3 |
| | | Omniquant | 48.29 | 75.42 | 77.92 | 77.80 | 75.59 | 45.20 | 80.41 | 46.62 | 70.17 | 66.38 | 5.02 |
| | | AWQ | 52.26 | 78.95 | 83.24 | 81.52 | 73.05 | 47.67 | 81.25 | 44.43 | 77.98 | 68.93 | 5.92 |
| | | QuaRot | 62.20 | 83.88 | 85.57 | 84.18 | 79.04 | 48.20 | 83.13 | 50.10 | 80.03 | 72.93 | 3.53 |
| | | SpinQuant | 62.03 | 84.97 | 85.11 | 84.06 | 78.30 | 47.00 | 83.90 | 49.85 | 80.90 | 72.90 | 3.49 |
| | | **OSTQuant** | 63.76 | 85.82 | 84.99 | 85.16 | 79.53 | 48.45 | 84.26 | 51.01 | 80.22 | 73.69 | 3.19 |
| | 4-4-16 | RTN | 27.47 | 25.88 | 37.83 | 26.26 | 0.00 | 27.20 | 51.63 | 35.26 | 49.33 | 31.21 | 9e3 |
| | | SmoothQuant | 25.60 | 34.47 | 50.46 | 32.48 | 1.98 | 30.00 | 54.24 | 33.83 | 48.93 | 34.67 | 2e2 |
| | | GPTQ | 25.77 | 26.09 | 43.64 | 26.42 | 0.00 | 27.40 | 52.01 | 32.55 | 49.33 | 31.47 | 4e4 |
| | | QuaRot | 50.60 | 73.65 | 77.46 | 77.83 | 71.96 | 43.20 | 78.13 | 45.29 | 71.90 | 65.56 | 6.35 |
| | | SpinQuant | 53.84 | 77.69 | 80.24 | 78.19 | 73.06 | 45.00 | 78.67 | 43.24 | 73.01 | 66.99 | 6.10 |
| | | **OSTQuant** | 61.84 | 84.56 | 84.14 | 82.47 | 77.08 | 46.07 | 83.38 | 50.23 | 80.13 | 72.21 | 3.97 |
| | 4-4-4 | RTN | 27.13 | 25.42 | 37.83 | 26.12 | 0.00 | 26.60 | 50.76 | 35.16 | 48.38 | 30.82 | 9e3 |
| | | SmoothQuant | 23.46 | 31.48 | 48.81 | 29.22 | 4.13 | 28.00 | 52.56 | 34.95 | 51.22 | 33.76 | 3e2 |
| | | GPTQ | 26.11 | 25.17 | 45.17 | 26.07 | 0.00 | 26.40 | 48.86 | 33.88 | 49.17 | 31.20 | 4e4 |
| | | QuaRot | 49.49 | 74.37 | 79.16 | 77.22 | 71.69 | 42.29 | 78.89 | 43.87 | 71.03 | 65.33 | 6.60 |
| | | SpinQuant | 51.88 | 76.39 | 80.98 | 76.50 | 71.43 | 43.46 | 79.27 | 44.17 | 72.69 | 66.31 | 6.24 |
| | | **OSTQuant** | 61.29 | 82.39 | 83.43 | 83.25 | 75.90 | 48.93 | 81.73 | 51.24 | 77.01 | 71.69 | 4.01 |

### A.6.2 Visualization Results

- Fig 10 illustrates the trends of activation QSUR and evaluation loss during the training process of the LLaMA3-8B model.
- Fig 11 and Fig 12 shows the activation distribution of different layers in LLaMA-2-7B and LLaMA-3-8B.
- Fig 13 shows the impact of Weight Outlier Minimization Initialization (WOMI) on weight distribution and quantization error of different layers in LLaMA-2-7B.
- Fig 14 presents visual comparisons of the activation distributions and quantization errors for QuaRot, SpinQuant, and OSTQuant on selected layers of LLaMA-2-7B.

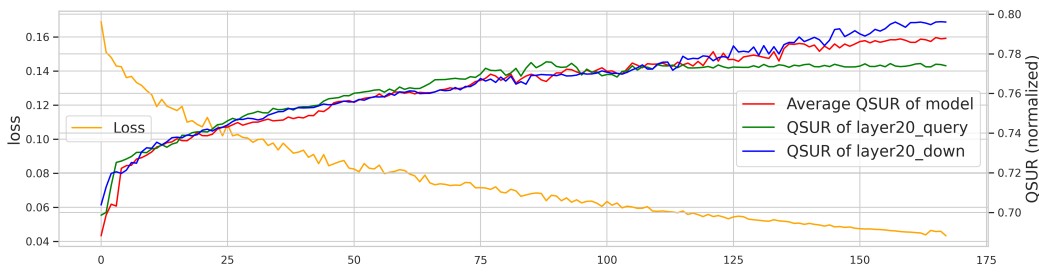

Figure 10: As the number of training iterations increases, the changes in the local and average QSURs, as well as the evaluation loss.

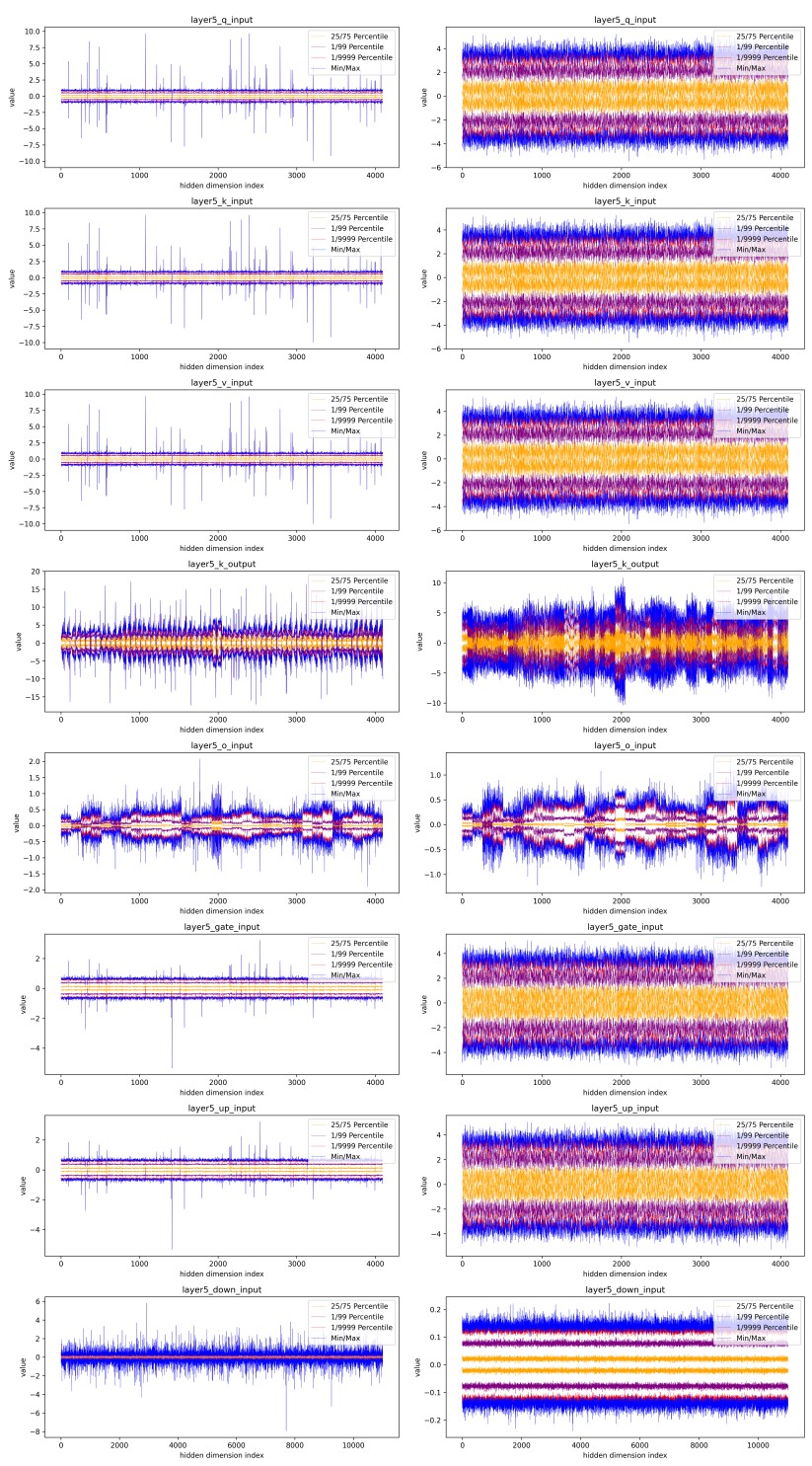

Figure 11: The activation distribution of different layers in LLaMA-2-7B before and after OSTQuant.

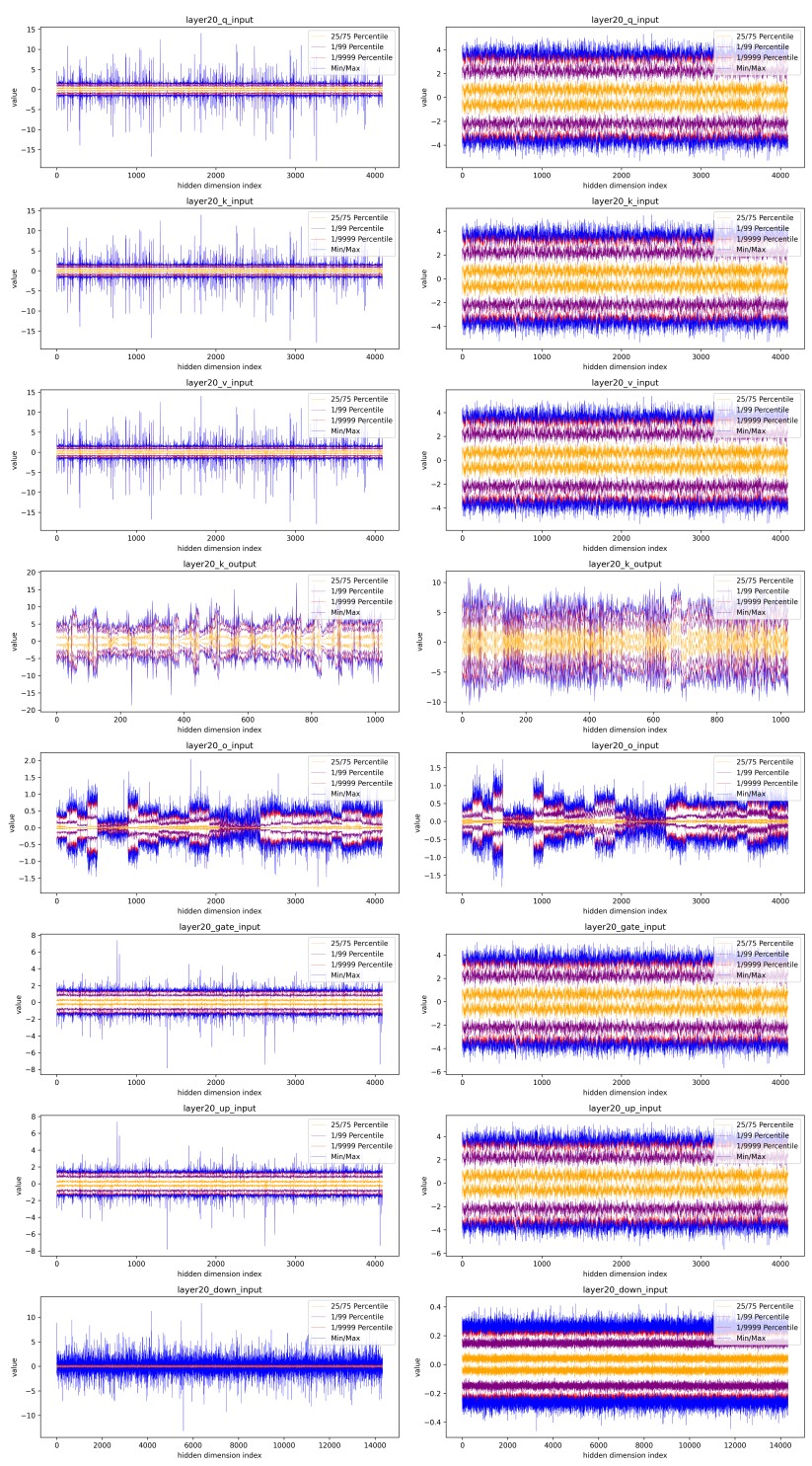

Figure 12: The activation distribution of different layers in LLaMA-3-8B before and after OSTQuant.

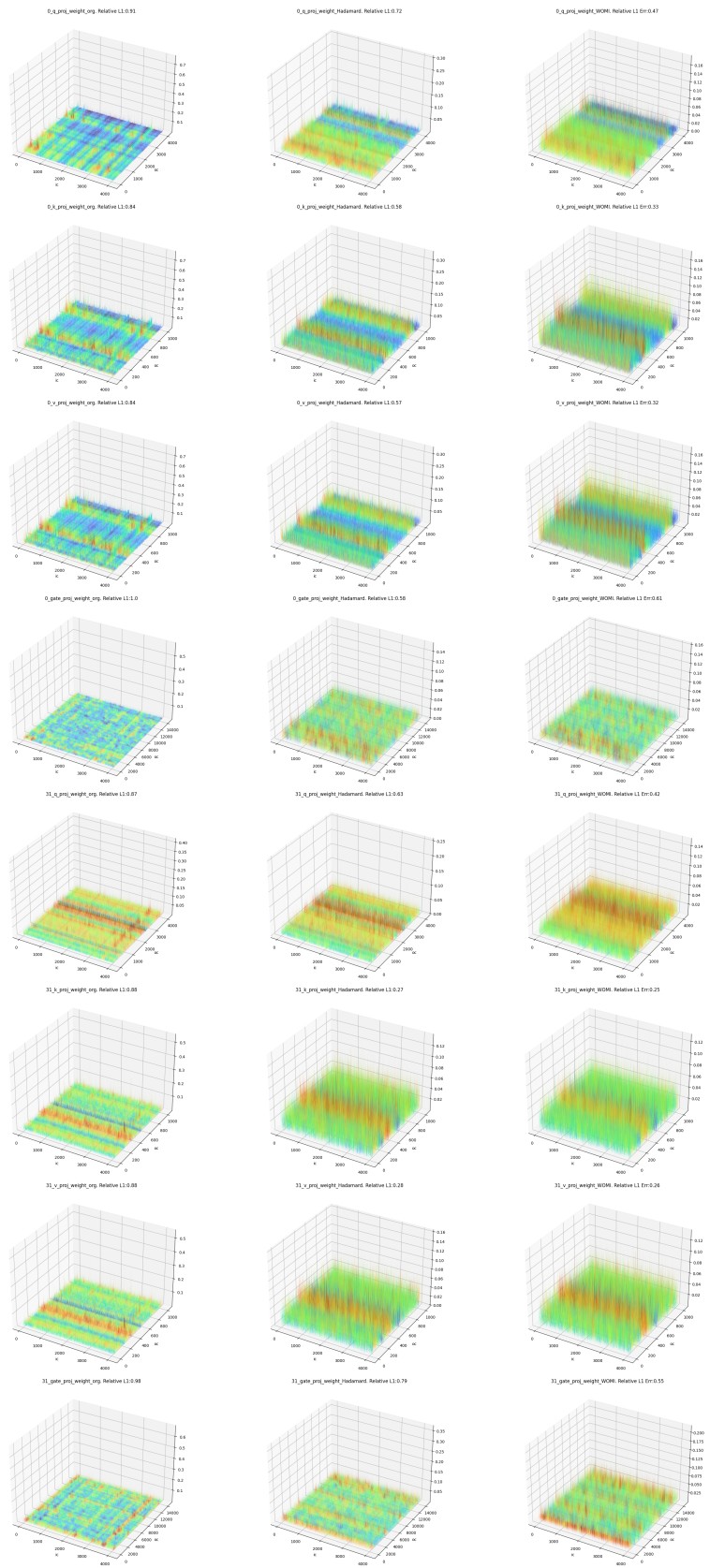

Figure 13: Visualizations comparing of the weight distribution and relative L1 error of LLaMA- 2-7B quantized with Ori (1st column), Hadamard transformed(2nd column), and WOMI transformed (3rd column), respectively.

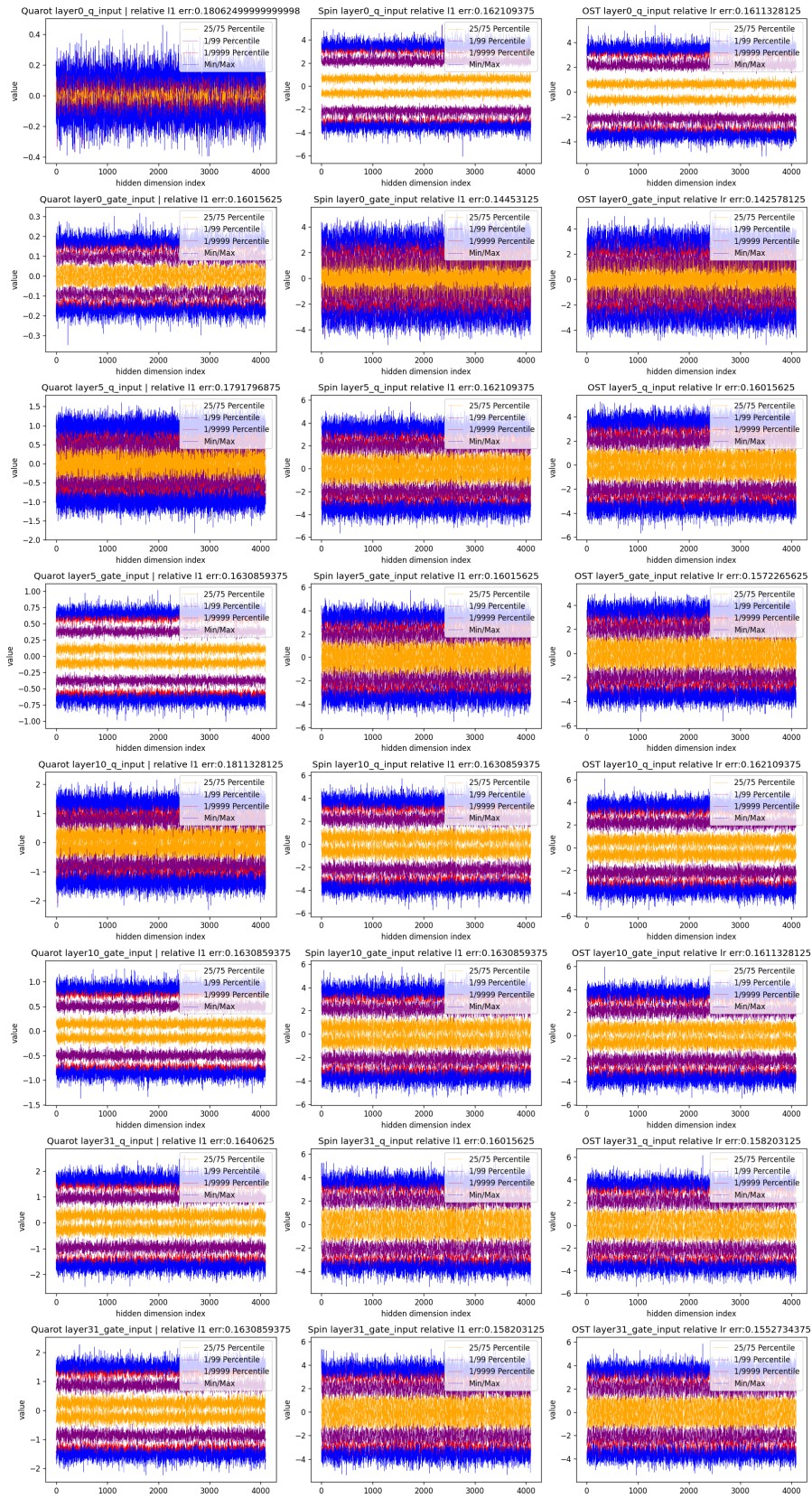

Figure 14: Visualizations comparing of the activation distribution and relative L1 error of LLaMA-2-7B quantized with QuaRot (1st column), SpinQuant (2nd column), and OSTQuant (3rd column), respectively. The relative L1 error here represents the difference between the activations before and after per-token quantization.

