# OpenReview forum: "OSTQuant: Refining Large Language Model Quantization with Orthogonal and Scaling Transformations for Better Distribution Fitting"
_ICLR.cc/2025/Conference — ICLR 2025 Poster_

### Official Review · Reviewer_NPKd · 2024-10-16

**Soundness:** 2
**Presentation:** 1
**Contribution:** 3
**Rating:** 5
**Confidence:** 5

**Summary:**

This paper introduces the Quantization Space Utilization Rate (QSUR) to assess data quantizability and guides the development of Orthogonal and Scaling Transformation-based Quantization (OSTQuant). OSTQuant optimizes weight and activation distributions in the quantization space and includes a KL-Top loss function to enhance semantic retention in post-training quantization. It outperforms state-of-the-art methods, maintaining 99.5% accuracy in simple configurations and significantly reducing performance gaps in more complex settings.

**Strengths:**

1. The paper introduces a new metric, QSUR, that links distribution characteristics with quantization space utilization, providing insight into the impact of transformation-based quantization approaches.
2. The proposed method for improving QSUR surpasses previous state-of-the-art techniques in performance.

**Weaknesses:**

1. Line 087 incorrectly identifies the two methods as block-wise reconstruction methods, which they are not.
2. In Equation (21), the direction of the inequality sign is incorrect. This error affects the subsequent proof, which also requires modification.
3. In Equation (4), the terms $\lambda_{min}$ and $\mathbf{q}\_{min}$ should be replaced with $\lambda\_{max}$ and $\mathbf{q}\_{max}$ respectively, in order to correctly derive $\mathbf{\mathit{v}}\_{min}^{org}$ for use in Equation (5). Please refer to the simple example shown in Figure 1(a). Corresponding changes are needed in the related sections of the paper.
4. The paper assumes that weights and activations are normally distributed. This assumption limits the generalizability of the results. The Laplace distribution, commonly used to model tensors, especially for outliers, is not considered. For further reference, see [this link](https://mobiusml.github.io/hqq_blog/).
5. Equation (29) appears to contain a clerical error. If this is not the case, a detailed explanation from the author would be beneficial.
6. The derivation in Section A.2.3 does not convincingly ensure that $|\mathbf{q}\_{max}|=d^{-\frac{1}{2}}$, which is necessary to obtain Equation (33).
7. Line 335 mentions "Figure X," which should be corrected to "Figure 5".
8. The use of the scaling vector $\mathbf{S}^i_{u|g}$ in Figure 5 under non-linear activation functions is not equivalent, and it is not employed in current studies. The author should explain the rationale behind using this vector.
9. The notations for vectors and matrices throughout the paper are inconsistent. For example, $S_{updown}$ in Table 4 versus $\mathbf{S}\_{u|d}^i$ in Figure 5. The paper should clarify the relationships among $\mathbf{S}\_{ffn}^i$, $\mathbf{S}\_{attn}^i$, and $\mathbf{S}^i$.
10. In Figure 5, $\mathbf{S}\_{qk}^i$ and $\mathbf{S}\_{ov}^i$ appear non-trainable. The paper does not provide an explanation for this choice.
11. Table 1 is not illustrated anywhere in the paper.
12. The paper does not include performance comparisons using larger Language Models, such as the 70B LLaMA-3, or investigate other model families and architectures, such as Mistral or Mixtal (MoE). Such comparisons could provide a broader evaluation of the proposed methods.

**Questions:**

1. Could the authors provide the speedup achieved during the decoding stage after implementing their proposed methods?
2. The paper should describe the initialization process for the scaling vectors used in the transformations.
3.  How does the model perform if only the initial matrix and vectors are employed to transform the model followed by naive quantization without training?
4. The performance improvement when compared with SpinQuant using a 4-4-4 quantization scheme appears to be less obvious in most cases than when using a 4-4-16 scheme. Can the authors explain why there is a disparity in performance improvements between these two configurations?
5. Regarding the activation distributions post-transformation, is there a noticeable difference among OSTQuant, SpinQuant, and QuaRot? It would be beneficial if the authors could include visualizations for both SpinQuant and QuaRot.

---

> ### Author Response · Authors · 2024-11-24
> **Response to Reviewer NPKd Part 1**
>
> We sincerely appreciate the effort you have put into reviewing our manuscript. We have carefully revised our article based on the issues you pointed out, including the typo errors in Weakness points 1, 2, and 5. We have tried our best to address all your concerns as follows.
>
> > In Equation (4), the terms $\lambda_{min}$ and $q_{min}$ should be replaced with $\lambda_{max}$ and $q_{max}$  respectively, in order to correctly derive $v_{min}^{org}$ vminorg for use in Equation (5).
>
> We use "max" and "min" to denote and distinguish two specific points, $\boldsymbol{x} _{max}$ and $\boldsymbol{x} _{min}$, in the  $\boldsymbol{X}$. Here, $\boldsymbol{x} _{min}$ refers to a point where one of its coordinate values is the smallest among all data, and $\boldsymbol{x} _{max}$ refers to a point where one of its coordinate values is the largest among all data. The eigenvalues and eigenvectors corresponding to $\boldsymbol{x} _{max}$ and $\boldsymbol{x} _{min}$ are denoted as $\lambda _{max}$, $\lambda _{min}$ and $\boldsymbol{q} _{max}$,$\boldsymbol{q} _{min}$, respectively.
>
> > The paper assumes that weights and activations are normally distributed. This assumption limits the generalizability of the results. The Laplace distribution, commonly used to model tensors, especially for outliers, is not considered. For further reference, see this link.
>
> We base our derivation on the normal distribution (Gaussian distribution) for the following reasons:
> 1. **The Hadamard transformation ensures weights and activations follow Gaussian distributions**
>
>     By employing WOMI initialization, the forward process is reformulated as $(\boldsymbol{XQ_wH}^T)(\boldsymbol{HQ_w}^T\boldsymbol{W})$. In this process, the forward propagation of weights and activations becomes $\boldsymbol{XQ_wH}^T$ and $\boldsymbol{W^TQ_wH^T}$, respectively, with both undergoing a Hadamard transformation. QuIP#[1] leverages the central limit theorem to demonstrate that the Hadamard transformation can convert the weight/activation distributions into Gaussian distributions.  Consequently, we focus on Gaussian distributions without extending our analysis to other distribution types.
> 2. **Gaussian Priors and Observations**
>
>     1. Normal distribution is a common assumption for network initialization and regularization:
>         1. Many studies and deep learning frameworks default to using Gaussian distributions for network parameter initialization. For example:
>             1. He et al. (2015), in their paper “Delving Deep into Rectifiers: Surpassing Human-Level Performance on ImageNet Classification”, proposed an initialization method based on Gaussian distributions.
>             2. Bengio et al. (2013), in “Learning Deep Architectures for AI”, also supported Gaussian-based initialization approaches.
>         2. Normalization layers in deep neural networks, such as BatchNorm and LayerNorm, further stabilize the mean and variance, making weights and activations closely approximate Gaussian distributions.
>     2. Impact of L2 regularization:
>
>         L2 regularization is a commonly used technique in the training of most large language models (LLMs). Its prior distribution is essentially Gaussian, strengthening the effectiveness of modeling weights and activations with normal distribution. The prior distribution corresponding to L1 regularization is the Laplace distribution.
>     3. **Empirical evidence:**
>
>         Figure 4(a) in our paper clearly demonstrates that the distributions of weights and activations in the LLaMA series models, across multiple dimensions, closely align with Gaussian distributions.
>
> Besides, regarding your mention of the Laplace distribution, we would like to clarify that, in studies such as HQQ[2], the Laplace distribution is used exclusively for modeling **quantized residuals(quantization error)** and is not applicable for modeling weights and activations.
>
> To more comprehensively address your concerns, we elaborate on the probability density function of a multivariate Laplace distribution $\boldsymbol{X}$, defined as:
> $$f(\boldsymbol{X} \mid \boldsymbol{\mu}, \boldsymbol{\Sigma}) = \frac{1}{(2\pi)^{n/2} |\boldsymbol{\Sigma}|^{1/2}} e^{-\frac{1}{2}(\boldsymbol{X} - \boldsymbol{\mu})},$$
> Its $\alpha$-quantiles also form a hyperellipsoid in high-dimensional space. OSTQuant can similarly flatten such distributions to improve QSUR. **Therefore, OSTQuant remains effective for Laplace distributions, as its core strategy involves leveraging equivalent transformations to optimize distribution flatness, thereby enhancing QSUR.**
>
> [1] Quip#: Even better LLM quantization with hadamard incoherence and lattice codebooks
> [2]HQQ: Half-Quadratic Quantization of Large Machine Learning Models

---

> ### Author Response · Authors · 2024-11-24
> **Response to Reviewer NPKd Part 2**
>
> > 1. The use of the scaling vector $\boldsymbol{S}_{u|g}^i$ in Figure 5 under non-linear activation functions is not equivalent, and it is not employed in current studies.
> > 2. The paper should clarify the relationships among $\boldsymbol{S} _{ffn}^i$, $\boldsymbol{S} _{attn}^i$, and $\boldsymbol{S}^i$.
> > 3. In Figure 5, $\boldsymbol{S} _{qk}^i$ and $\boldsymbol{S} _{ov}^i$ appear non-trainable.
>
> Thanks for your suggestion, we have revised Figure 5 to improve its clarity and added Figure 9 to illustrate the strategy for full quantization.
>
> In OSTQuant, we introduce two distinct strategies for two different quantization modes: traditional LLM quantization and full quantization (where all activations in the Transformer Block are quantized). **The number and types of learnable transformations differ between these two modes.** To clarify these strategies, we have included illustrative diagrams in the revised version. Specifically, Figure 5 represents the strategy for traditional quantization, while Figure 9 illustrates the strategy for full quantization. We will now explain each strategy in detail.
>
> ---
> As shown in Figure 5 of Section 4.1 in revision, the trainable orthogonal matrices are represented using diamonds, including the following:
> - Global $\boldsymbol{R}_{res}$, which applies a unitary orthogonal transformation to activations along all residual paths.
> - $n\cdot \boldsymbol{R^i}_{ov}$, Correspond to each block , where $n$represents the number of heads in  Attention module.
>
> The trainable scaling matrixs include:
> - $\boldsymbol{S^i_{attn}}$, used to smooth inter-channel differences in weights and activations in the QKV projection layers of the Attention Block.
> -  $\boldsymbol{S^i_{ffn}}$, used  to smooth inter-channel differences in the weights and activations of the up and gate projection layers of the MLP Block.
> - $\boldsymbol{S^i_{ov}}$, to smooth inter-channel differences in the weights and activations of the output projection layer in the Attention Block.
> - $\boldsymbol{S^i_{u|d}}$, used to smooth inter-channel differences in the weights and activations of the down projection layer in the MLP Block.
>
> The various learnable scaling and orthogonal transformations are depicted in Figure 5. Online Hadamard transformations are represented by gray diamond icons. By introducing these learnable parameters, we establish pairs of orthogonal and scaling transformations for each linear layer’s activations and weights to improve their QSUR and reduce quantization errors. Moreover, after optimization, all scaling transformations, along with orthogonal transformations other than $\boldsymbol{R}_{qk}$, can be fused into the corresponding weights, avoiding additional parameters and computational overhead.
>
> ---
> Figure 9 introduces OSTQuant’s novel strategy for full quantization. Full quantization involves quantizing all activations within each Transformer Block to low precision. This reduces memory transfer overhead for all activations and fully leverages efficient low-precision computational units.
>
> As shown in Figure 9, OSTQuant incorporates additional equivalence transformations to modify the distributions of all node input and output activations. Unlike the traditional quantization method in Figure 5, the fully quantized design adds more learnable transformations, especially around the ROPE and SiLU layers. Specifically:
>
> 1. **ROPE Handling**: We treat ROPE as a lightweight GEMM layer and construct a weight matrix with shape (token,head_dim,head_dim) based on its principle. Pre-ROPE and post-ROPE transformation pairs are introduced as follows:
>     1. Pre-ROPE transformations are based on the fact that ROPE and the preceding linear layer can be viewed as consecutive matrix multiplications along the head_dim. The corresponding transformation pairs are represented in figure by $\boldsymbol{S_q^iR^i_{pre}}$ and $\boldsymbol{S_k^iR^i_{pre}}$.
>     2. Post-ROPE transformations rely on the Attention computation formula $\boldsymbol{Attn = Q @ K^\top}$. The corresponding transformation pairs are represented in figure by $\boldsymbol{S^i_{post}R^i_{post}}$
> 2. **Smoothing Activation Discrepancies of SiLU**: Inspired by the smoothing method for SwiGLU introduced in I-LLM[1], we decompose SiLU as $\mathrm{SiLU}(\boldsymbol{X}) = \boldsymbol{X} \cdot \sigma(\boldsymbol{X})$ and use equivalence such as $$\boldsymbol{
>         X_1 \cdot X_2 \cdot \sigma(X_1) = (X_1 \cdot S) \cdot (S_2 \cdot \frac{1}{S}) \cdot \sigma((X_1 \cdot S) \cdot \frac{1}{S})
>         }$$ to alleviate inter-channel discrepancies of activation before and after SiLU. The corresponding transformation is represented in figure by $\boldsymbol{S_{u|g}}$.
>
> [1]I-LLM: Efficient Integer-Only Inference for Fully-Quantized Low-Bit Large Language Models

---

> ### Author Response · Authors · 2024-11-24
> **Response to Reviewer NPKd Part 3**
>
> > The derivation in Section A.2.3 does not convincingly ensure that $|\boldsymbol{q}_{max}|=d^{\frac{-1}{2}}$, which is necessary to obtain Equation (33)
>
> We would like to clarify: in the derivation in Section A.2.3, it is not necessary to satisfy $|\boldsymbol{q}_{max}| = d^{-\frac{1}{2}}$, nor is this condition a prerequisite for deriving Equation 33.
>
> Specifically,  for a single sample following Gaussian distribution, **we transform the covariance matrix of the original distribution into an identity matrix using the transformation $\boldsymbol{T = \Lambda^{-\frac{1}{2}} Q^\top}$** from Equation 30. In this case, all eigenvalues become equal, and the eigenvectors are normalized unit vectors. Under such conditions, according to Equation 7, we can derive Equation 33.
>
> It seems there might have been a misunderstanding between the formulas in Appendix A.2.2 and Appendix A.2.3. The goal of A.2.3 is to identify the optimal transformation matrix, whereas the expression $|\boldsymbol{q}_{max}| = d^{-\frac{1}{2}}$ appears in A2.2, which focuses on finding the best orthogonal matrix. These two objectives are largely independent and not directly related.
>
> > Table 1 is not illustrated anywhere in the paper.
>
> In section 4.2, line 368 (line **371** in the revised version), we have referenced this table to demonstrate the effectiveness of KL-Top loss function.
>
> > Could the authors provide the speedup achieved during the decoding stage after implementing their proposed methods?
>
> In Table R2 of General Response, we complement the speedup and memory saving in decoding stage.
>
> > The paper should describe the initialization process for the scaling vectors used in the transformations.
>
> Thank you for your suggestion. We have revised the manuscript to include the following: "For all scaling matrices, we initialize them as identity matrices." To avoid a cumbersome and imprecise calibration phase, we initialize the scaling matrices as identity matrices. The optimal scaling matrices are then obtained through global optimization.
>
> > How does the model perform if only the initial matrix and vectors are employed to transform the model followed by naive quantization without training?
>
> We've analyzed the impact of employing only the initial matrix and vectors through visualizations of the weights and performance analysis of the model.
>
> In the appendix A.3.1 of the revised manuscript, Figure 7 demonstrates the impact of Weight Outlier Minimization Initialization (WOMI) on weight distribution and quantization error. Additionally, Table R5-2 below compares the effectiveness of WOMI and Hadamard initialization in QuaRot[2].
>
> As shown in Figure 7, the original weight distribution exhibits significant variations across input and output channels. While Hadamard initialization reduces inter-channel differences, noticeable spikes still persist. WOMI, leveraging the Hadamard matrix and the covariance matrix of the weight distribution, further smooths these inter-channel differences, effectively reducing the quantization space and relative quantization error.
>
> Table R5-2 below highlights the performance of LLaMA-2-7B and LLaMA-3-8B models initialized with WOMI and random Hadamard matrices. WOMI achieves lower perplexity and higher few-shot accuracy under both W4A4KV4 and W4A16KV16 configurations, showcasing its effectiveness. Interestingly, WOMI demonstrates greater performance improvements in W4-only quantization settings compared to W4A4KV4. This is likely due to WOMI’s superior capability in minimizing weight quantization errors, which is especially critical in W4-only configurations.
>
> Table R5-2: Comparison of the impact of WOMI initialization and Hadamard initialization introduced by QuaRot on the performance of quantized models.
> | **Model**  | **Quant Setting** | **Method** | **Zero-Shot^9** | **Wiki PPL** |
> |:----------:|:-----------------:|:----------:|:---------------:|:------------:|
> |            | FP                | -          | 65.21           | 5.47         |
> | LLaMA-2-7B | W4A16KV16         | Hadamard   | 63.32           | 5.62         |
> |            | W4A16KV16         | WOMI       | 63.45           | 5.59         |
> |            | W4A4KV4           | Hadamard   | 61.47           | 6.11         |
> |            | W4A4KV4           | WOMI       | 61.52           | 6.09         |
> |            | FP                | -          | 68.09           | 6.14         |
> | LLaMA-3-8B | W4A16KV16         | Hadamard   | 67.27           | 6.53         |
> |            | W4A16KV16         | WOMI       | 67.41           | 6.53         |
> |            | W4A4KV4           | Hadamard   | 61.38           | 8.18         |
> |            | W4A4KV4           | WOMI       | 61.40           | 8.17         |

---

> ### Author Response · Authors · 2024-11-24
> **Response to Reviewer NPKd Part 4**
>
> > The performance improvement when compared with SpinQuant using a 4-4-4 quantization scheme appears to be less obvious in most cases than when using a 4-4-16 scheme. Can the authors explain why there is a disparity in performance improvements between these two configurations?
>
> This is a particularly noteworthy phenomenon, which may be attributed to **SpinQuant’s own anomalous fluctuations about performance:**
>
> In our experiments, we utilized SpinQuant’s official evaluation code and pre-trained checkpoints. Interestingly, for LLaMA-2-7B and LLaMA-2-13B, the performance of the W4A4KV16 configuration was slightly lower than that of W4A4KV4, which contradicts intuitive expectations. We also observed certain issues about accuracy reproduction in the SpinQuant official repository, but no definitive explanations have been provided so far. This anomalous baseline behavior could have contributed to the observed significance of the performance differences.
>
> When compared to QuaRot, the improvements observed on 4-4-16 and 4-4-4 configurations align with expectations. For instance, on LLaMA-2-7B, W4A4KV4 achieves a 2-point improvement in zero-shot^9, while W4A4KV4 sees a 1.7-point increase.
>
> > Regarding the activation distributions post-transformation, is there a noticeable difference among OSTQuant, SpinQuant[1], and QuaRot[2]? It would be beneficial if the authors could include visualizations for both SpinQuant and QuaRot.
>
> Figure 14 in the revised manuscript illustrates the activation distribution and quantization error of LLaMA-2-7B when quantized using QuaRot (1st column), SpinQuant (2nd column), and OSTQuant (3rd column), respectively. Although QuaRot gets the smallest quantization range (vertical axis), it exhibits the highest relative L1 error between pre- and post-quantization activations. This is evident from the pronounced fluctuations in the differences between the min-max percentile and the 99th percentile within each row, highlighting QuaRot's less stable handling of outliers.
>
> In contrast, both SpinQuant and OSTQuant obtain more consistent activation distributions as they incorporate optimization during training. Notably, OSTQuant, with its greater optimization capacity, achieves significantly lower quantization error compared to SpinQuant, along with a more convergent and stable quantization range.
>
> [1]Quarot: Outlier-free 4-bit inference in rotated llms
> [2]SpinQuant--LLM quantization with learned rotations

---

> > ### Comment · Reviewer_NPKd · 2024-11-25
> >
> > Thanks for the reply. However, I still have lots of concerns here:
> > * The performance in the paper is bizarre. For example, lots of results show incremental improvement compared with SpinQuant. However, it demonstrates a great enhancement on LLaMA3-70B, which is not included in SpinQuant.
> > * Only considering the method in the paper, too many tricks with the incremental results, e.g., w4a16kv16 and that with LLaMA2-70B and LLaMA2-30B.
> >
> > By the way, I still think the current version's presentation is unclear and not easy to follow. For example, in Line 343, "Figure X" needs to be fixed and methods and theories seem to be somewhat disconnected (the authors simply and brutally learn the transformation, which I think needs to be explored more to align with their theory).
> >
> > Overall, I acknowledge the contribution of the theoretical analysis here, yet the methods and results seem incremental across many models and bit widths. Therefore, I keep the initial score.

---

> ### Author Response · Authors · 2024-11-25
> **There are several important points we would like to clarify.**
>
> Dear reviewer NPKd:
> > The performance in the paper is bizarre. For example, lots of results show incremental improvement compared with SpinQuant. However, it demonstrates a great enhancement on LLaMA3-70B, which is not included in SpinQuant.
>
> We have open-sourced our code at https://anonymous.4open.science/r/OSTQuant-05F2, ensuring that all our results are fully reproducible. Additionally, the SpinQuant results we reported are totally and originally based on their open source code open-source code (https://github.com/facebookresearch/SpinQuant). **If you have any concerns or questions regarding our code or results, we are more than willing to provide more details and make any corrections.**
>
> It is worth noting that SpinQuant reported the results of LLama3-70B in version V2 (https://arxiv.org/pdf/2405.16406v2) but subsequently removed them in version V3 (https://arxiv.org/pdf/2405.16406). Additionally, there have been concerns regarding the reproducibility of accuracy in the official SpinQuant repository (https://github.com/facebookresearch/SpinQuant/issues/11), yet no clear or definitive explanations have been provided to address these issues.
>
> > Only considering the method in the paper, too many tricks with the incremental results, e.g., w4a16kv16 and that with LLaMA2-70B and LLaMA2-30B.
>
> Regarding the concern about "too many tricks" in "Only considering the method in the paper, too many tricks with the incremental results," we would like to clarify that the methods we employed are strictly the three core contributions outlined in Section 4 METHODOLOGY: Equivalent Transformation Pair, Weight Outlier Minimization Initialization (WOMI), and KL-TOP LOSS. **No additional tricks were used beyond these methods.** **Furthermore, all experimental results were conducted under fair and consistent settings (e.g., W4A4KV16, W4A16KV16, etc.)**.  For transparency and fairness, we also reported the results of applying KL-TOP LOSS on SpinQuant.
>
> We sincerely thank you for pointing out the typos. We will carefully correct them and further improve the readability of our paper.

---

> > ### Comment · Reviewer_NPKd · 2024-11-27
> >
> > Thanks for the reply. I am willing to reconsider the score.

---

> > > ### Author Response · Authors · 2024-11-28
> > >
> > > Dear Reviewer NPKd
> > >
> > > Thank you for your constructive comments and valuable suggestions.
> > > We greatly appreciate the time and effort you have dedicated to reviewing our manuscript. Your feedback has been instrumental in improving its overall quality and clarity.
> > > We are pleased to address your questions and sincerely value your recognition of our work.
> > >
> > > Best, OSTQuant Authors

---

> > > ### Author Response · Authors · 2024-11-29
> > >
> > > Dear Reviewer NPKd
> > >
> > > Thank you again for your thoughtful review. We believe that, after incorporating your suggestions as well as those from the other reviewers, the latest version represents a substantial improvement over the initial one you evaluated. Therefore, we kindly ask if you could take some time to review our improvements and provide a re-evaluation.
> > >
> > > Best Regards,
> > > OSTQuant Authors

---

> > > > ### Author Response · Authors · 2024-12-02
> > > > **Gentle Reminder and Invitation for Further Feedback**
> > > >
> > > > Dear Reviewer NPKd,
> > > >
> > > > We hope this message finds you well. We would like to express our sincere gratitude for your thoughtful feedback and for your willingness to reconsider your score on our paper. As the deadline for review updates is approaching, we would like to check if you have any additional feedback or if there are any further clarifications we can provide to assist in your reassessment.
> > > >
> > > > Your insights have been invaluable in improving our work, and we truly appreciate the time and effort you’ve invested. Please let us know if there’s anything else we can do.
> > > >
> > > > Thank you again for your consideration.
> > > >
> > > > Warm regards,
> > > >
> > > > The Authors of OSTQuant

---

### Official Review · Reviewer_uSGo · 2024-10-30

**Soundness:** 2
**Presentation:** 3
**Contribution:** 2
**Rating:** 6
**Confidence:** 3

**Summary:**

This paper presents a mathematical framework to evaluate the effectiveness of smooth and rotation transformations in the context of outlier suppression during LLM quantization. It introduces a metric, QSUR, to assess the efficacy of various transformations and derives a closed-form solution for the optimal transformation. Based on QSUR, the paper proposes OSTQuant, a method that optimizes the combination of smooth and orthogonal transformations globally and employs KL-Top loss to mitigate noise during PTQ optimization.

There are some concerns regarding the methodology and assumptions in the paper.

(1). Quantization Assumption: The paper assumes the volume of the quantization space $V_{S_{X}}$ as that of a cube when dimension $d=3$, which implies per-tensor quantization. However, the standard practice is to use per-token quantization for activations and per-channel quantization for weights. The volume of per-channel quantization is typically a cuboid when dimension $d = 3$. The authors should clarify if their assumptions are based on these scenarios and how their methodology applies to these more common quantization schemes.

(2). Mean Vector Neglect: In line 231, the paper states," we neglect the mean vector $µ$, so $λ_{max} = λ_{min} = λ$". The causal relationship between neglecting the mean vector and the equality of the maximum and minimum eigenvalues is not evident and requires further explanation.

(3). Optimization of Transformations: The paper optimizes the pair of smooth and orthogonal transformations, with the transformation initialized by the closed-form optimal derivation. It is unclear why it is necessary to further optimize a transformation that is already designed to reach maximum QSUR. Additionally, the paper lacks an ablation study on the impact of different orthogonal initialization methods.

(4). Correlation of QSUR and Model Performance: Figure 3 shows a positive correlation between QSUR and the performance of the quantized model. However, it is not demonstrated whether a higher QSUR value consistently leads to higher performance in quantized models. The paper should provide a more rigorous validation to establish this relationship across various datasets and architectures.

(5). Hadamard Transformation: The paper uses online Hadamard transformation for query and key. It is unclear whether the Hadamard transformation can be optimized as suggested in Figure 5.

**Strengths:**

The paper provides a mathematical framework that attempts to understand the role of smooth and rotation transformations in quantization, which is a novel approach. The introduction of QSUR offers potential insights into the quantization process, and the closed-form solution for the optimal transformation could serve as a theoretical foundation for further research. The integration of orthogonal and scaling transformations within OSTQuant could address issues related to outliers in activations and weights, which is a significant contribution.

**Weaknesses:**

A major concern is that, despite deriving a closed-form solution for the optimal transformation based on QSUR, the paper opts to use learnable transformation parameters initialized by the optimal transformation instead of directly applying the theoretically derived optimal transformation. This choice should be justified.

While the paper establishes a positive correlation between QSUR and the performance of the quantized model, it is unclear whether a higher QSUR necessarily indicates a superior transformation. There is a lack of rigorous validation demonstrating that transformations with higher QSUR values consistently lead to better model performance across various datasets and architectures.

The hypervolume $V_{S_{X}}$ of the quantization space $S_{X}$ is calculated using per-tensor quantization, whereas common practice typically involves per-token quantization for activations and per-channel quantization for weights. The authors need to clarify how QSUR applies to these more conventional quantization schemes and validate their metric in these contexts. Without this clarification, the applicability and robustness of QSUR are questionable.

In summary, while the paper provides interesting theoretical contributions, there are substantial issues with the practical application of the derived optimal transformations and the validity of QSUR as a reliable indicator of transformation quality.

**Questions:**

(1). Why is the closed-form optimal transformation derived from QSUR not directly applied in the model?

(2). Does a higher QSUR value guarantee a better transformation in terms of model performance?

(3). Which of the transformation matrices mentioned in the paper are Hadamard matrices and which are unit orthogonal matrices, and are Hadamard matrices trainable as well?

(4). Should softmax be applied before or after the top-k selection, and what is the rationale behind this choice?

---

> ### Author Response · Authors · 2024-11-24
> **Response to Reviewer uSGo Part 1**
>
> We sincerely thank you for your valuable time and efforts in reviewing our manuscript, and we tried our best to address all your concerns as follows.
>
> > A major concern is that, despite deriving a closed-form solution for the optimal transformation based on QSUR, the paper opts to use learnable transformation parameters initialized by the optimal transformation instead of directly applying the theoretically derived optimal transformation. This choice should be justified.
> > While the paper establishes a positive correlation between QSUR and the performance of the quantized model, it is unclear whether a higher QSUR necessarily indicates a superior transformation. There is a lack of rigorous validation demonstrating that transformations with higher QSUR values consistently lead to better model performance across various datasets and architectures.
>
> Actually, we've derived two closed-form solutions to enhance QSUR:
>
> 1. Equation 8 $\boldsymbol{T = \Lambda^{-\frac{1}{2}} Q^\top}$ serves as the theoretical foundation for inspiring us to propose orthogonal ($\boldsymbol{Q}^\top$) and scaling ($\boldsymbol{\Lambda}^{-\frac{1}{2}}$) equivalent transformation pairs in OSTQuant. While QSUR is inversely proportional to quantization error within a single layer, its inherent nature as a local metric makes it unlikely to achieve a globally optimal solution. Therefore, we further propose KL-top loss function to globally optimize these transformation pairs, enabling adaptive improvements to the QSUR of both activations and weights across all layers.
>
> 2. Equation 27 $\boldsymbol{T} = \boldsymbol{H}\boldsymbol{Q^\top} $, where $\boldsymbol{H}$ is the normalized Hadamard matrix and $\boldsymbol{Q}$ is composed of all unit eigenvectors of the weights. This equation provides a critical foundation for proposed Weight Outlier Minimization Initialization (WOMI), which offers the following advantages:
>
>     - It eliminates outliers in the weights and enhances the QSUR of the weights ($\boldsymbol{W}' = \boldsymbol{HQ}^T\boldsymbol{W}$), as demonstrated in Figure 13 and Appendix A.2.2.
>     - It enhances the initial QUSR of activations ($\boldsymbol{X}' = \boldsymbol{XQH}^T)$ because $\boldsymbol{Q}\boldsymbol{H}^T$ serves as a random orthogonal matrix followed by a Hadamard transform $\boldsymbol{H}^\top$, which can reduce inter-channel variance.
>     - It ensures quantization stability (i.e., higher initial QSUR, as shown in Figure 10) and accelerates convergence of subsequent optimization (about 150 iterations).
>
>     We also compared the differences between the Hadamard initialization and the WOMI initialization.  As shown in Table 7, WOMI achieves higher accuracy, especially under the W4-only quantization configuration.
>
> Figure 3 in the manuscript demonstrates the impact of different quantization methods on the quantized performance and overall QSUR of the LLaMA series models. To improve clarity, we have reorganized the data from Figure 3 and related tables into below Table R4-1. The results presented in Table R4-1 indicate a positive correlation between QSUR and the quantized performance of LLMs. To address your concern, we include Figure 10 in the appendix to further demonstrate that the evaluation loss aligns with local QSURs. **Thus, we can confirm that transformations with higher QSUR values consistently lead to better model performance.**
>
> Table R4-1: Zero-Shot$^9$ precision retention (under W4A4KV4 quantization) and normalized QSUR are evaluated for LLaMA variants across different quantization methods.
> | **Model**  | **Metric**         | **RTN** | **SmoothQuant** | **SpinQuant** | **OSTQuant** |
> |:----------:|:------------------:|:-------:|:---------------:|:-------------:|:------------:|
> | LlaMA1 7B  | QSUR               | 0.04    | 0.12            | 0.75          | 0.77         |
> |            | 0-Shot^9 retention | 0.51    | 0.52            | 0.95          | 0.97         |
> | LLaMA1 13B | QSUR               | 0.00    | 0.02            | 0.79          | 0.79         |
> |            | 0-Shot^9 retention | 0.47    | 0.50            | 0.97          | 0.98         |
> | LLaMA2 7B  | QSUR               | 0.01    | 0.04            | 0.76          | 0.78         |
> |            | 0-Shot^9 retention | 0.50    | 0.49            | 0.95          | 0.97         |
> | LLaMA2 13B | QSUR               | 0.01    | 0.03            | 0.77          | 0.80         |
> |            | 0-Shot^9 retention | 0.46    | 0.49            | 0.95          | 0.97         |
> | LlaMA3 8B  | QSUR               | 0.01    | 0.03            | 0.74          | 0.76         |
> |            | 0-Shot^9 retention | 0.49    | 0.48            | 0.94          | 0.96         |

---

> ### Author Response · Authors · 2024-11-24
> **Response to Reviewer uSGo Part 2**
>
> > The hypervolume $V_{S_X}$ of the quantization space $S_X$ is calculated using per-tensor quantization, whereas common practice typically involves per-token quantization for activations and per-channel quantization for weights. The authors need to clarify how QSUR applies to these more conventional quantization schemes and validate their metric in these contexts. Without this clarification, the applicability and robustness of QSUR are questionable.
>
> In our derivation, we employ the per-tensor form of quantization computation. This approach is chosen because the quantization operations of all linear layers can be reformulated into the per-tensor format.
>
> Specifically, consider the forward propagation process of a linear layer, expressed as:
> $$\boldsymbol{Y = XW},$$
> where $\boldsymbol{X}$ uses per-token quantization, and $\boldsymbol{W}$ adopts per-channel quantization.
>
> When calculating the QSUR, we normalize activations and weights to unify them into the per-tensor form. Concretely:
>     - Activation $\boldsymbol{X}$ are normalized along the token dimension.
>     - Weight $\boldsymbol{W}$ are normalized along the channel dimension.
>
> The normalization can be expressed as:
> $$\boldsymbol{X} = \boldsymbol{m \frac{X}{m}}, \quad \boldsymbol{W} = \boldsymbol{\frac{W}{n} n},$$
> where $\boldsymbol{m}$ is a vector consisting of the maximum values of each activation token, and $\boldsymbol{n}$ is a vector consisting of the maximum values of each weight channel.
>
> After normalization, the forward propagation formula becomes:
> $$\boldsymbol{Y} = \boldsymbol{m \hat{X} \hat{W} n},$$
> where $\boldsymbol{\hat{X}}$ and $\boldsymbol{\hat{W}}$ are the normalized per-token activations and per-channel weights, respectively. At this stage, **the quantization behavior of $\boldsymbol{\hat{X} \hat{W}}$ is equivalent to per-tensor quantization.**
>
> Thus, our method naturally adapts to common quantization schemes, such as per-token activation quantization and per-channel weight quantization. Adopting the per-tensor form during derivation does not affect the applicability of QSUR.
>
> > Which of the transformation matrices mentioned in the paper are Hadamard matrices and which are unit orthogonal matrices, and are Hadamard matrices trainable as well?
>
> As shown in Figure 5, all matrices denoted by the symbol $\boldsymbol{R}$ are unitary orthogonal matrices, and those denoted by $\boldsymbol{S}$ are diagonal matrices. Specifically, $\boldsymbol{R} _ {qk}$ and $\boldsymbol{R}_{d|n}$ represent the fast Hadamard transform. To enhance your understanding of each equivalent transformation in OSTQuant, we have redrawn Figure 5 in the revised manuscript.
>
> In fact, with an appropriate design, the Hadamard transforms following the Q and K projections can be learned and become imperceptible after optimization. This has been elaborated in the design of OSTQuant for full quantization(quantizing all activations within each Transformer Block to low bits).  Figure 9 in revision provides an intuitive depiction of this design, and Appendix A.5.1 explains the implementation details.
>
> > Should softmax be applied before or after the top-k selection, and what is the rationale behind this choice?
>
> Softmax should be applied before the Top-k selection, and only a subset of the Softmax results should be used.
>
> Specifically, we first use the indices $\boldsymbol{I}$ and the corresponding probabilities $\boldsymbol{P^I}$ of the top-k categories from the floating-point model outputs as the labels. After the quantized model generates logits, we compute its Softmax results $\boldsymbol{Q}$, then extract the probabilities $\boldsymbol{Q^I}$ for the top-k categories based on the indices $\boldsymbol{I}$. Finally, the KL divergence loss is calculated between $\boldsymbol{P^I}$ and $\boldsymbol{Q^I}$.
>
> The reason for applying Softmax before the Top-k selection is to ensure that the probability computation takes into account all logits. **Ignoring part of the logits prior to the Softmax calculation would skew the probability distribution and compromise accuracy.** By first computing the global Softmax and then focusing on the probabilities of the specific top-k categories, this approach avoids errors caused by disregarding certain logits. It also effectively reduces noise interference and mitigates overfitting risks on small calibration data.

---

> > ### Comment · Reviewer_uSGo · 2024-11-26
> >
> > Thanks a lot for the detailed response from the authors, which has addressed my concerns. I will increase my rate.

---

> > > ### Author Response · Authors · 2024-11-27
> > >
> > > Dear Reviewer uSGo,
> > >
> > > Thank you for your constructive comments and valuable suggestions.
> > >
> > > We are glad to address all your questions and sincerely appreciate your recognition.
> > >
> > > Best, Authors

---

### Official Review · Reviewer_n22x · 2024-11-03

**Soundness:** 3
**Presentation:** 3
**Contribution:** 3
**Rating:** 6
**Confidence:** 4

**Summary:**

The paper introduces QSUR and proposes OSTQuant that improves quantization performance by globally optimizing multiple equivalent transformation pairs in LLMs with KL-Top loss. Each transformation consists of a scaling transformation and an orthogonal transformation.

**Strengths:**

This paper mathematically derives a new metric QSUR to quantify the utilization of the quantization space, which is experimentally found to be positively correlated with the accuracy of the quantized model. Based on QSUR, the authors define the optimal transform matrix, which inspires them to combine the scaling and orthogonal transformation to minimize the quantization space. By incorporating scaling transformation and KL-Top loss, OSTQuant achieves SOTA results across various quantization settings while maintaining a low calibration cost.

**Weaknesses:**

While having mathematically derived the form of an optimal transform matrix, OSTQuant only applies weight outlier minimization initialization to the orthogonal transformation while leaving scaling transformation behind, incurring a small gap between the theoretical analysis and the method. Besides, it would be helpful to show the improvements of applying weight outlier minimization initialization compared with Hadamard matrix. It would also be beneficial to include experiment results on larger models (e.g. 70B) and decoding speedup.

**Questions:**

1. Can the authors showcase the effect of initializing the transform matrix with the theoretically derived form?

2. The experiment results on larger models such as LLaMA-2-70B and LLaMA-3-70B are missed in the paper, can the authors provide more experiment results on these models?

3. The decoding speedup is also missed in the paper. It would be helpful to include latency analyses on the decoding latency.

---

> ### Author Response · Authors · 2024-11-24
> **Response to Reviewer n22x Part 1**
>
> We sincerely thank you for your valuable time and efforts in reviewing our manuscript, and we tried our best to address all your concerns as follows.
>
> > While having mathematically derived the form of an optimal transform matrix, OSTQuant only applies weight outlier minimization initialization to the orthogonal transformation while leaving scaling transformation behind, incurring a small gap between the theoretical analysis and the method.
>
> Actually, we've derived two closed-form solutions to enhance QSUR:
>
> 1. Equation 8 $\boldsymbol{T = \Lambda^{-\frac{1}{2}} Q^\top}$ serves as the theoretical foundation for inspiring us to propose orthogonal ($\boldsymbol{Q}^\top$) and scaling ($\boldsymbol{\Lambda}^{-\frac{1}{2}}$) equivalent transformation pairs in OSTQuant. While QSUR is inversely proportional to quantization error within a single layer, its inherent nature as a local metric makes it unlikely to achieve a globally optimal solution. Therefore, we further propose KL-top loss function to globally optimize these transformation pairs, enabling adaptive improvements to the QSUR of both activations and weights across all layers.
> 2. Equation 27 $\boldsymbol{T} = \boldsymbol{H}\boldsymbol{Q^\top} $, where $\boldsymbol{H}$ is the normalized Hadamard matrix and $\boldsymbol{Q}$ is composed of all unit eigenvectors of the weights. This equation provides a critical foundation for proposed Weight Outlier Minimization Initialization (WOMI), which offers the following advantages:
>     - It eliminates outliers in the weights and enhances the QSUR of the weights ($\boldsymbol{W}' = \boldsymbol{HQ}^T\boldsymbol{W}$), as demonstrated in Figure 13 and Appendix A.2.2.
>     - It enhances the initial QUSR of activations ($\boldsymbol{X}' = \boldsymbol{XQH}^T)$ because $\boldsymbol{Q}\boldsymbol{H}^T$ serves as a random orthogonal matrix followed by a Hadamard transform $\boldsymbol{H}^\top$, which can reduce inter-channel variance.
>     - It ensures quantization stability (i.e., higher initial QSUR, as shown in Figure 10) and accelerates convergence of subsequent optimization (about 150 iterations).
>
>     We also compared the differences between the Hadamard initialization and the WOMI initialization.  As shown in Table 7 in the revised manuscript, WOMI achieves higher accuracy, especially under the W4-only quantization configuration.

---

> ### Author Response · Authors · 2024-11-24
> **Response to Reviewer n22x Part 2**
>
> > Can the authors showcase the effect of initializing the transform matrix with the theoretically derived form?
>
> In Appendix A.3.1 of the revised manuscript, Figure 7 in the revised manuscript demonstrates the impact of Weight Outlier Minimization Initialization (WOMI) on LLaMA-2-7B. Additionally, Table R3-1 below(or Table 7 in the revised manuscript) compares the effectiveness of WOMI and Hadamard initialization.
>
> As shown in Figure 7, the original weight distribution exhibits significant variations across input and output channels. While Hadamard initialization reduces inter-channel differences, noticeable spikes still persist. WOMI, by incorporating both the Hadamard matrix and the covariance matrix of the weight distribution, further smooths these differences, effectively reducing the quantization space and quantization error.
>
> Table R3-1 below highlights the performance of LLaMA-2-7B and LLaMA-3-8B models initialized with WOMI and random Hadamard matrices like QuaRot[1]. WOMI achieves lower perplexity and higher few-shot accuracy under both W4A4KV4 and W4A16KV16 configurations, showcasing its effectiveness. Notably, WOMI shows greater performance improvements in W4-only quantization settings compared to W4A4KV4. This is likely due to WOMI’s superior ability to minimize weight quantization errors, which is particularly important in W4-only configurations.
>
> Table R3-1: Comparison of the impact of WOMI initialization and Hadamard initialization on the performance of quantized models.
> | **Model**  | **Quant Setting** | **Method** | **Zero-Shot^9** | **Wiki PPL** |
> |:----------:|:-----------------:|:----------:|:---------------:|:------------:|
> |            | FP                | -          | 65.21           | 5.47         |
> | LLaMA-2-7B | W4A16KV16         | Hadamard   | 63.32           | 5.62         |
> |            | W4A16KV16         | WOMI       | 63.45           | 5.59         |
> |            | W4A4KV4           | Hadamard   | 61.47           | 6.11         |
> |            | W4A4KV4           | WOMI       | 61.52           | 6.09         |
> |            | FP                | -          | 68.09           | 6.14         |
> | LLaMA-3-8B | W4A16KV16         | Hadamard   | 67.27           | 6.53         |
> |            | W4A16KV16         | WOMI       | 67.41           | 6.48         |
> |            | W4A4KV4           | Hadamard   | 61.38           | 8.18         |
> |            | W4A4KV4           | WOMI       | 61.40           | 8.17         |
>
> > The experiment results on larger models such as LLaMA-2-70B and LLaMA-3-70B are missed in the paper, can the authors provide more experiment results on these models?
>
> As shown in Table R0 of the General Response (or Table 14 in the revised manuscript), we evaluated OSTQuant’s performance on LLaMA-3-70B across different quantization configurations including W4A16KV16, W4A4KV16, and W4A4KV4. Experiments show that OSTQuant achieves state-of-the-art results under all configurations, demonstrating its effectiveness and strong generalization capability on larger-scale language models. Results of LLaMA-2-70B can also be found in Table 14 of revised manuscript.
>
> > The decoding speedup is also missed in the paper. It would be helpful to include latency analyses on the decoding latency.
>
> As shown in Table R2 of General Response, we complement the speedup and memory saving in decoding stage.
>
>
> [1]Quarot: Outlier-free 4-bit inference in rotated llms

---

> > ### Comment · Reviewer_n22x · 2024-11-25
> >
> > Thanks for the authors' response, which has addressed most of my concerns. I maintain my original positive score.

---

> > > ### Author Response · Authors · 2024-11-26
> > >
> > > Dear Reviewer n22x
> > >
> > > Thank you for your constructive comments and valuable suggestions. We sincerely appreciate your recognition.
> > >
> > > Best regards,
> > > Authors of OSTQuant

---

### Official Review · Reviewer_dw3S · 2024-11-04

**Soundness:** 3
**Presentation:** 2
**Contribution:** 3
**Rating:** 6
**Confidence:** 5

**Summary:**

This work proposes a PTQ technique that reportedly achieves SOTA results on LLMs.
The technique employs an equivalent transformation before quantizer which combines smoothing and rotation in recent work, together with optimization of a top-K KLD loss and a weight-statistics-cognizant initialization of rotation.
A local heuristic metric QSUR is introduced to assess the efficiency of quantization.
Experiments and ablation studies were carried out with popular LLMs.

**Strengths:**

+ Well-motivated idea.
+ Adequate presentation of background and recent related work.
+ Comprehensive experiments.

**Weaknesses:**

- Certain ablation studies are still missing that are necessary to establish the contributions from different components of the proposed technique (see below).
- Quality of writing could use further improvement.  For example, there seems to be a mislabeling of $\Lambda$ as $A$ in equations (9) and (10); certain figure legends, such as Fig. 6, are too tiny to read.

**Questions:**

* One major question I have is on the relative contributions by the 3 new ingredients in the proposed method: the equivalence pair reparameterization, the top-K KL loss, and the initialization.  The authors did conduct ablation study on the special initialization, but there seems to be a lack of ablation on the loss.  In other words, it is not clear how much the differential results in Table 2 comparing against other methods are due to the combination of smoothing and rotation, and due to the top-K KLD loss--what would be the result if one applies SpinQuant but optimizing the top-K KLD?
* Another major question that I have is the role of QSUR.  The authors provided theoretic results on the effect of linear transformation on this local metric assuming Gaussianity--this is good.  But in practice, QSUR is not an objective function for direct optimization in conducting OSTQuant, but rather, OSTQuant minimizes a global KLD loss.  This begs the question: how do the global loss align with local QSURs?  During the course of OSTQuant optimization, if one measure QSUR across various layers in the network, do they all monotonically grow as the global loss monotonically decreases?

**Details Of Ethics Concerns:**

None.

---

> ### Author Response · Authors · 2024-11-24
> **Response to Reviewer dw3S Part 1**
>
> We sincerely appreciate the time and effort you have dedicated to reviewing our manuscript. In the revised version, we have carefully reviewed and corrected global writing and formatting issues and made adjustments to the equations and figures based on your suggestions. We have made every effort to address all your concerns, as detailed below.
>
> > There seems to be a lack of ablation on the loss. In other words, it is not clear how much the differential results in Table 2 comparing against other methods are due to the combination of smoothing and rotation, and due to the top-K KLD loss--what would be the result if one applies SpinQuant but optimizing the top-K KLD?
>
> Thank you for your suggestion. We provide Table R2-1 in the comment below and have updated to Table 8 in the revised manuscript. As shown, even with the KL-Top loss function, using SpinQuant alone still fails to significantly close the gap with OSTQuant and may even lead to some performance degradation. We believe the reasons for this gap are as follows:
>
> - Better theoretical support. Compared to SpinQuant, which focuses solely on optimizing, the orthogonal and scaling transformations in OSTQuant have been proven to offer a higher greater potential for improving QSUR.
>
> - Broader Parameter Optimization Space. The parameter optimization space of SpinQuant is limited, consisting of a global learnable rotation transformation $\boldsymbol{R_1}$ and a block-level rotation transformation $\boldsymbol{R_2}$.  Building on this, OSTQuant not only learns multiple scaling transformations within each block but also learns head-specific OS transformation pairs between Value projection and Out projection.
>
> Table R2-1: Performance of SpinQuant and OSTQuant with the introduction of the KL-Top loss function on 9 zero-shot dataset tasks and the Perplexity changes on Wikitext2.
>
> | **Model** | **Method**         | **ARC-c** | **ARC-e** | **BoolQ** | **HellaS** | **Lam.** | **OBQA** | **PIQA** | **SIQA** | **WinoG.** | **Avg.**    | **Wiki2 PPL** |
> |-----------|--------------------|-----------|-----------|-----------|------------|----------|----------|----------|----------|------------|-------------|---------------|
> |           | RTN                | 23.72     | 30.56     | 46.18     | 29.83      | 2.70     | 28.60    | 52.45    | 34.39    | 50.20      | 33.18       | 704.34        |
> |           | SpinQuant          | 46.33     | 73.57     | 76.15     | 75.43      | 71.40    | 41.40    | 79.16    | 44.68    | 68.75      | 64.10       | 7.35          |
> | LLaMA3-8B | SpinQuant + KL-Top | 47.29     | 73.95     | 75.82     | 75.64      | 71.40    | 41.58    | 78.16    | 44.38    | 68.45      | 64.07       | 7.54          |
> |           | OSTQuant           | 49.26     | 76.68     | 78.25     | 76.18      | 70.48    | 43.19    | 77.85    | 45.18    | 69.13      | 65.13       | 6.80          |
> |           | OSTQuant + KL-Top  | 49.32     | 76.73     | 78.87     | 76.01      | 70.77    | 43.20    | 78.51    | 45.70    | 69.22      | 65.37       | 7.29          |
> |           | RTN                | 27.22     | 27.06     | 50.83     | 27.34      | 0.93     | 25.8     | 49.51    | 34.85    | 50.51      | 32.67       | nan           |
> | LLaMA2-7B | SpinQuant          | 40.44     | 71.08     | 74.4      | 73.51      | 70.66    | 41.8     | 76.88    | 43.5     | 65.82      | 62.01       | 5.956         |
> |           | SpinQuant + KL-Top | 40.76     | 71.29     | 74.61     | 73.08      | 70.19    | 40.94    | 76.32    | 43.85    | 67.78      | 62.09       | 6.16          |
> |           | OSTQuant           | 42.41     | 69.87     | 75.07     | 72.9       | 70.21    | 40.87    | 78.16    | 44.16    | 68.4       | 62.45       | 5.38          |
> |           | OSTQuant + KL-Top  | 44.62     | 72.69     | 75.41     | 73.27      | 70.21    | 41       | 78.13    | 44.42    | 68.27      | 63.11       | 5.935         |

---

> ### Author Response · Authors · 2024-11-24
> **Response to Reviewer dw3S Part 2**
>
> > But in practice, QSUR is not an objective function for direct optimization in conducting OSTQuant, but rather, OSTQuant minimizes a global KLD loss. This begs the question: how do the global loss align with local QSURs?
>
> QSUR is a local metric designed to evaluate the quantization adaptability of a set of data $\boldsymbol{X}$ from a distributional perspective. We investigate the influence of linear transformation on QSUR and derive a closed-form solution  $\boldsymbol{T = \Lambda^{-\frac{1}{2}} Q^\top}$(Equation 8)  to maximum the QSUR of a single layer. This serves as the theoretical foundation for inspiring us to propose orthogonal ($\boldsymbol{Q^\top}$) and scaling ($\boldsymbol{\Lambda^{-\frac{1}{2}}}$) equivalent transformation pairs to improve the quantifiability of data distribution.
>
> While QSUR is inversely proportional to quantization error within a single layer, its inherent nature as a local metric makes it unlikely to achieve a globally optimal solution.Therefore, we further propose KL-top loss function to globally optimize these transformation pairs , enabling adaptive improvements to the QSUR of both activations and weights across all layers. In our original version, figure 3 illustrates that the evaluation metrics align with QSUR. To address your concern, we include Figure 10 in the appendix to further demonstrate that the evaluation loss aligns with local QSURs.
>
>
> > During the course of OSTQuant optimization, if one measure QSUR across various layers in the network, do they all monotonically grow as the global loss monotonically decreases?
>
> QSUR plays a critical role in determining model performance and is positively correlated with it. While QSUR is inversely proportional to quantization error within a single layer, its inherent nature as a local metric makes it unlikely to achieve a globally optimal solution. Therefore, we further propose KL-top loss function to globally optimize these transformation pairs, enabling adaptive improvements to the QSUR of both activations and weights across all layers.
>
> Figure 3 demonstrates the impact of different quantization methods on the quantized performance and overall QSUR of the LLaMA series models. To improve clarity, we have reorganized the data from Figure 3 and related tables into Table R2-2. The results presented in Table R2-2 indicate a positive correlation between QSUR and the quantized performance of LLMs. Notably, the OSTQuant method achieves the highest performance in both QSUR and quantization metrics.
>
> We observed that the QSUR of most layers increases as the global loss decreases, while a few initially increase and then plateau. In our revised manuscript, we add Figure 10 in the appendix to further demonstrate that the evaluation loss aligns with QSURs.
>
> Table R2-2: Zero-Shot$^9$ precision retention (under W4A4KV4 quantization) and normalized QSUR are evaluated for LLaMA variants across different quantization methods.
> | **Model**  | **Metric**         | **RTN** | **SmoothQuant** | **SpinQuant** | **OSTQuant** |
> |:----------:|:------------------:|:-------:|:---------------:|:-------------:|:------------:|
> | LlaMA1 7B  | QSUR               | 0.04    | 0.12            | 0.75          | 0.77         |
> |            | 0-Shot^9 retention | 0.51    | 0.52            | 0.95          | 0.97         |
> | LLaMA1 13B | QSUR               | 0.00    | 0.02            | 0.79          | 0.79         |
> |            | 0-Shot^9 retention | 0.47    | 0.50            | 0.97          | 0.98         |
> | LLaMA2 7B  | QSUR               | 0.01    | 0.04            | 0.76          | 0.78         |
> |            | 0-Shot^9 retention | 0.50    | 0.49            | 0.95          | 0.97         |
> | LLaMA2 13B | QSUR               | 0.01    | 0.03            | 0.77          | 0.80         |
> |            | 0-Shot^9 retention | 0.46    | 0.49            | 0.95          | 0.97         |
> | LlaMA3 8B  | QSUR               | 0.01    | 0.03            | 0.74          | 0.76         |
> |            | 0-Shot^9 retention | 0.49    | 0.48            | 0.94          | 0.96         |

---

> ### Author Response · Authors · 2024-12-02
> **Warm Appreciation and Invitation for Further Feedback**
>
> Dear Reviewer dw3S,
>
> We hope this message finds you well. We would like to sincerely thank you for your thoughtful review and valuable feedback on our paper. Your positive assessment and constructive comments have been instrumental in helping us improve our work.
>
> In response to your concerns, we have:
> * **Conducted Additional Ablation Studies:** We have included new ablation studies to better illustrate the contributions of the different components of our proposed technique. Specifically, we examined the impact of the KL-Top loss function when applied to both SpinQuant and OSTQuant. The results, now presented in Table 8 of the revised manuscript, show that even with the KL-Top loss, SpinQuant does not achieve the same level of performance as OSTQuant, highlighting the effectiveness of our proposed methods.
> * **Clarified the Role of QSUR:**  We have expanded our discussion on how the global loss aligns with local QSURs. We included additional figures (e.g., Figure 10 in the appendix) to demonstrate that as the global loss decreases during optimization, QSURs across various layers generally increase, indicating a positive correlation between QSUR and model performance.
>
> As we approach the final day for review updates, we would like to check if you have any additional feedback or if there are further clarifications we can provide. Your insights are invaluable to us, and we truly appreciate the time and effort you’ve invested in reviewing our work.
>
> If you find that our revisions have satisfactorily addressed your concerns, we would be grateful if you could consider reflecting this in your final assessment.
>
> Thank you again for your valuable contributions to our research.
>
> Warm regards,
>
> The Authors of OSTQuant

---

### Official Review · Reviewer_347N · 2024-11-06

**Soundness:** 4
**Presentation:** 3
**Contribution:** 4
**Rating:** 8
**Confidence:** 2

**Summary:**

This paper proposes the OSTQuant approach for LLM post-training quantization (PTQ), which assigns learnable Orthogonal and Scaling Transformations to each fully connected (FC) layer in LLMs. It also introduces the Quantization Space Utilization Rate (QSUR) as a metric to evaluate the quantizability of transformed data. Additionally, the KL-Top loss function is introduced to address the small calibration set issue.

**Strengths:**

- The proposed QSUR provides a quantitative index to evaluate the effectiveness of the transformations. It motivates the OSTQuant method, and the results demonstrate its high effectiveness.

- The paper achieves significantly improved results for 4-bit activation, weight, and KV cache quantization.

- The paper is mostly well-written. I particularly like Figure 1 which provides a clear illustration of the concepts.

**Weaknesses:**

- Since this is a PTQ method, it would be beneficial to demonstrate its effectiveness on larger models, such as a 70B parameter model. Additionally, showing a scaling curve with model size, memory cost, computation cost, and quantization bit would help users understand the overall Pareto frontier between cost and quality.

- Some parts of the paper could be clearer, particularly in detailing certain aspects. For example, while Figure 1 is helpful, it took some time to understand the meaning of the grey dots and ovals.  Similarly, Table 3 lacks details about the inference system and why only prefill speedup is mentioned. Providing more context and explanations would improve the paper’s readability and accessibility.

**Questions:**

- It would be good to see the speedup results on hardware with FP4 support, like Blackwell architecture, to fully show the potential.

---

> ### Author Response · Authors · 2024-11-24
> **Response to Reviewer 347N**
>
> We sincerely express our gratitude for your valuable time and efforts spent in reviewing our manuscript. We wholeheartedly agree that demonstrating the effectiveness of OSTQuant on larger models (see Table 14 of revision) and presenting comprehensive scaling curves (see Appendix A.4 and Figure 8 of revision) would greatly enhance the understanding of our proposed method. We have made every effort to address all your concerns as follows.
>
> > It would be beneficial to demonstrate its effectiveness on larger models, such as a 70B parameter model. Additionally, showing a scaling curve with model size, memory cost, computation cost, and quantization bit would help users understand the overall Pareto frontier between cost and quality.
>
> We validate the effectiveness of OSTQuant on the LLaMA-3-70B and LLaMA-2-70B  and include the results into the revised revision.
>
> As shown in Table R0 of the General Response (or Table 14 in the revised manuscript), we evaluated OSTQuant’s performance on LLaMA-3-70B across different quantization configurations including W4A16KV16, W4A4KV16, and W4A4KV4. Experiments show that OSTQuant achieves state-of-the-art results under all configurations, demonstrating its effectiveness and strong generalization capability on larger-scale language models. Results of LLaMA-2-70B can also be found in Table 14 of revised manuscript.
>
> In the revised manuscript, Figure 8 illustrates the performance and model size of the Llama series under various quantization settings. It can be observed that the w4 setting achieves the best trade-off.
>
> > Some parts of the paper could be clearer, particularly in detailing certain aspects. For example, Figure 1 is helpful, it took some time to understand the meaning of the grey dots and ovals. Similarly, Table 3 lacks details about the inference system.
>
> In the revised manuscript, we provide the detailed explanation of the meaning of the gray points and the ellipses in Figure 1. The ellipses represent the distribution space of the data $\boldsymbol{X}$ that conforms to a two-dimensional distribution and the squares represent the quantization space required for quantizing this distribution. The boundaries of the quantization space are determined by the maximum and minimum values of the batch data. The gray points within the square denote specific quantization points within the quantization space. When the number of quantization points included within the ellipse increases, it indicates a higher utilization of the quantization space by distribution, corresponding to a higher QSUR.
>
> > Table 3 lacks details about the inference system and why only prefill speedup is mentioned.
>
> The inference environment of Table 3 features an Intel(R) Xeon(R) Gold 5317 CPU and an Nvidia 3090 GPU. The 4-bit matrix multiplication kernel was implemented using cutlass of nvidia, and the self-attention mechanism utilized PyTorch’s native SDPA (scaled dot product attention) function. All tests were conducted 500 times, with the median value taken as the final result.
>
> In Table R2 of General Response, we complement the speedup and memory saving in decoding stage.
>
> > It would be good to see the speedup results on hardware with FP4 support, like Blackwell architecture, to fully show the potential.
>
> We appreciate the reviewers’ constructive feedback and for highlighting potential directions for extending OSTQuant. At present, hardware that supports FP4 (such as the Blackwell architecture) is still limited, making related experimental conditions somewhat constrained. Nevertheless, we firmly believe that FP4 represents one of the key future trends in quantization technology, and we will continue to monitor and explore FP4-related quantization research. In the future, we plan to conduct further experiments on FP4-supported hardware to fully demonstrate the potential and applicability of OSTQuant.

---

> ### Author Response · Authors · 2024-12-02
> **Heartfelt Gratitude for Your Encouraging Feedback**
>
> Dear Reviewer 347N,
>
> We hope this message finds you well. We would like to express our sincere gratitude for your thoughtful and encouraging review of our paper. Your positive assessment and kind words mean a great deal to us.
>
> We are delighted that you found value in our proposed QSUR metric and the OSTQuant method. Your recognition of our work motivates us to continue our research in this area.
>
> Thank you again for your time and for sharing your valuable insights. We deeply appreciate your support.
>
> Warm regards,
>
> The Authors of OSTQuant

---

### Author Response · Authors · 2024-11-24
**General Response: revision update notice**

Dear Reviewers,
We sincerely thank all the reviewers for their efforts and time spent on carefully reviewing our work, OSTQuant. The paper has been moderately revised based on the valuable suggestions provided. We have thoroughly considered all the concerns and questions raised by the reviewers, and our reflections on these issues have guided the revisions made to the manuscript. These changes have enhanced the comprehensiveness of the paper and improved the readability of OSTQuant. All modifications in the manuscript are highlighted in blue in the revised version. Below, we have summarized the main revisions and provided explanations for each:

- We have open-sourced our code at https://anonymous.4open.science/r/OSTQuant-05F2, ensuring that all our results are fully reproducible.
- The performance of OSTQuant on the 70B model.
  - Add the performance results of OSTQuant on LLaMA-2-70B and LLaMA-3-70B to demonstrate the effectiveness of OSTQuant, as shown in Table 14.  [thanks for the suggestion from reviewers 347N, n22x, and NPKd.]
- Supplementation and adjustments to figures and tables.
  - Redraw Figure 5 with clearer annotations to better illustrate our method. Additionally, we present the design of OSTQuant for full quantization in schematic Figure 9.  [addressing concerns about the effects of different matrices from reviewers usGo and NPKd]
  - Provide detailed explanations of the ellipses and gray dots in Figure 1. [thanks for the suggestion from reviewer 347N.]
  - Add activation distribution plots after quantization for SpinQuant, QuaRot, and OSTQuant, along with the corresponding relative quantization errors (see Figure 14). [suggestion from reviewer NPKd]
  - Add Figure 8 to illustrate the accuracy performance and memory consumption of the LLaMA series models under different quantization bitwidths. [suggestion from reviewer 347N.]
  - Address some typographical errors and make adjustments to the dimensions of figures.  [thanks to the suggestions and comments from reviewers dw3S and NPKd.]
- Additional ablation experiments.
  - Add the impact of WOMI on weight distribution and performance, with a detailed analysis in Appendix 3.1. Visual results have also been added in Figure 13. [suggestion from reviewer n22x]
  - Add experiments combining KL-Top Loss and Spinquant, along with a detailed analysis (see Appendix 3.2). [suggestion from reviewer n22x]
- Additional efficiency analysis.
  - Provide the acceleration performance and memory savings of OSTQuant during the decoding stage for language models of different sizes (see Table 10).  [suggestion from reviewer 347N,n22x,NPKd]

We sincerely thank everyone for their valuable suggestions on OSTQuant, which will help make it even better. If you still have concerns about specific aspects, please feel free to discuss them with us at any time.

Sincerely

The Authors

---

> ### Author Response · Authors · 2024-11-24
> **General Response: Performance of OSTQuant on 70B Models and Decoding Stage Speedup Part1**
>
> In response to the concerns raised by Reviewer 347N, n22x, and NPKd regarding OSTQuant's performance on larger language models, we present its evaluation results on LLaMA-3-70B in Table R1 below. Consistent with the original paper, we include results for three quantization configurations: W4A4KV4, W4A4KV16, and W4A16KV16. These results encompass perplexity (PPL) performance on Wikitext-2 and accuracy across nine zero-shot datasets. Additionally, the results for LLaMA-2-70B are provided in Table 14 of the revised manuscript.
>
> Table R1: Comparison of perplexity on WikiText 2 and accuracy on 9 zero-shot tasks of OSTQuant and other methods on the LLaMA-3-70B model.
>
> | **W-A-KV** |   **Method**   | **ARC-c** | **ARC-e** | **BoolQ** | **HellaS** | **Lam.** | **OBQA** | **PIQA** | **SIQA** | **WinoG.** | **Avg** | **Wiki2 PPL** |
> | :--------: | :------------: | :-------: | :-------: | :-------: | :--------: | :------: | :------: | :------: | :------: | :--------: | :-----: | :-----------: |
> |  16 16 16  | Full Precision |   64.42   |   85.98   |   85.14   |   84.95    |  79.47   |  48.46   |  84.39   |  50.82   |   80.66    |  73.81  |     2.86      |
> |  4 16 16   |      RTN       |   26.28   |   25.55   |   37.83   |   26.36    |   0.00   |  29.00   |  50.98   |  34.70   |   49.64    |  31.15  |      1e5      |
> |            |  SmoothQuant   |   51.88   |   77.53   |   80.09   |   80.47    |  73.16   |  46.60   |  80.58   |  45.29   |   75.85    |  67.94  |     6.70      |
> |            |      GPTQ      |   25.77   |   25.29   |   37.83   |   26.36    |   0.12   |  28.40   |  51.74   |  34.90   |   52.64    |  31.45  |      9e3      |
> |            |      AWQ       |   52.26   |   78.95   |   83.24   |   81.52    |  73.05   |  47.67   |  81.25   |  44.43   |   77.98    |  68.92  |     5.92      |
> |            |     QuaRot     |   62.20   |   83.88   | **85.57** |   84.18    |  79.04   |  48.20   |  83.13   |  50.10   |   80.03    |  72.93  |     3.53      |
> |            |   SpinQuant    |   62.03   |   84.97   |   85.11   |   84.06    |   78.3   |  47.00   |   83.9   |  49.85   | **80.90**  |  72.90  |     3.49      |
> |            |      Ours      | **63.76** | **85.82** |   84.99   | **85.16**  |**79.53** |**48.45** |**84.26** |**51.01** |   80.22    |**73.69**|   **3.19**    |
> |            |                |           |           |           |            |          |          |          |          |            |         |               |
> |   4 4 16   |      RTN       |   27.47   |   25.88   |   37.83   |   26.26    |   0.00   |  27.20   |  51.63   |  35.26   |   49.33    |  31.21  |      8e3      |
> |            |  SmoothQuant   |   25.60   |   34.47   |   50.46   |   32.48    |   1.98   |  30.00   |  54.24   |  33.83   |   48.93    |  34.67  |      2e2      |
> |            |      GPTQ      |   25.77   |   26.09   |   43.64   |   26.42    |   0.00   |  27.40   |  52.01   |  32.55   |   49.33    |  31.47  |      4e4      |
> |            |     QuaRot     |   50.60   |   73.65   |   77.46   |   77.83    |  71.96   |  43.20   |  78.13   |  45.29   |   71.90    |  65.56  |     6.35      |
> |            |   SpinQuant    |   53.84   |   77.69   |   80.24   |   78.19    |  73.06   |  45.00   |  78.67   |  43.24   |   73.01    |  66.99  |     6.10      |
> |            |      Ours      | **61.84** | **84.56** | **84.14** | **82.47**  |**77.08** |**46.07** |**83.38** |**50.23** | **80.13**  |**72.21**|   **3.97**    |
> |            |                |           |           |           |            |          |          |          |          |            |         |               |
> |   4 4 4    |      RTN       |   27.13   |   25.42   |   37.83   |   26.12    |   0.00   |  26.60   |  50.76   |  35.16   |   48.38    |  30.82  |      8e3      |
> |            |  SmoothQuant   |   23.46   |   31.48   |   48.81   |   29.22    |   4.13   |  28.00   |  52.56   |  34.95   |   51.22    |  33.76  |      3e2      |
> |            |      GPTQ      |   26.11   |   25.17   |   45.17   |   26.07    |   0.00   |  26.40   |  48.86   |  33.88   |   49.17    |  31.20  |      4e4      |
> |            |     QuaRot     |   49.49   |   74.37   |   79.16   |   77.22    |  71.69   |  42.29   |  78.89   |  43.87   |   71.03    |  65.33  |      6.6      |
> |            |   SpinQuant    |   51.88   |   76.39   |   80.98   |   76.50    |  71.43   |  43.46   |  79.27   |  44.17   |   72.69    |  66.31  |     6.24      |
> |            |      Ours      | **61.29** | **82.39** | **83.43** | **83.25**  |**75.90** |**48.93** |**81.73** |**51.2**  | **77.01**  |**71.69**|   **4.01**    |

---

> ### Author Response · Authors · 2024-11-24
> **General Response: Performance of OSTQuant on 70B Models and Decoding Stage Speedup Part2**
>
> In response to the suggestions from Reviewer 347N, n22x, and NPKd, we report the speedup of the decoding stage in Table R2 below. The experiments were conducted on a single Nvidia A6000 GPU paired with an Intel® Xeon® Gold 5317 CPU. To evaluate generation speed in more practical scenarios, we first have the model perform a prefill operation with a sequence of 2048 tokens. Subsequently, we execute 2048 decoding steps for speed testing. This process is repeated 100 times with a batch size of 1, and the median value is used as the final result. Due to memory limitations, the floating-point inference speed for the 30B and 70B models on a single GPU could not be obtained. Nevertheless, the results demonstrate that, with nearly lossless accuracy, even a single A6000 GPU can run LLaMA-3-70B at a speed of 15 tokens per second after quantization.
>
> Table R2 : Comparison of generation speed and memory usage before and after 4-bit quantization in the decoding stage. All tests were conducted on an NVIDIA A6000 GPU.
> | **Model**   | **decoder speed (tokens/sec)** |           |          | **Memory use (GB)** |           |                |
> | :---------: | :----------------------------: | :-------: | :------: | :-----------------: | :-------: | :------------: |
> | -           | FP                             | Quantized | Speed up | FP                  | Quantized | Memory saveing |
> | LLaMA-2-7B  | 47.32                          |   89.4    |  1.89x   | 13.94               |   4.32    |     3.23x      |
> | LLaMA-3-8B  | 38.33                          |   77.71   |  2.03x   | 15.83               |   5.88    |     2.69x      |
> | LLaMA-2-13B | 23.7                           |   55.35   |  2.34x   | 23.7                |    8.5    |     2.79x      |
> | LLaMA-30B   | OOM                            |   30.49   |    -     | OOM                 |   18.19   |       -        |
> | LLaMA-3-70B | OOM                            |   14.68   |    -     | OOM                 |   38.41   |       -        |

---

### Meta-Review · Area_Chair_cSXg · 2024-12-20

**Metareview:**

This paper presents OSTQuant, a novel post-training quantization technique for large language models that introduces orthogonal and scaling transformations for each fully connected layer. The method also proposes the Quantization Space Utilization Rate (QSUR) as a metric for evaluating transformation effectiveness, alongside a KL-Top loss to address small calibration datasets. The results demonstrate significant improvements in quantization, particularly with 4-bit activations, weights, and KV cache.

The authors have effectively addressed the reviewers' comments and provided strong experimental evidence to support their approach. The method is well-motivated and demonstrates clear advances in quantization performance. While some minor clarifications and additional experiments could enhance the paper further, the overall contribution is strong.

The area chair recommend acceptance of the paper, with the suggestion that the authors make final version modifications to improve clarity and address any remaining reviewer concerns.

**Additional Comments On Reviewer Discussion:**

During the rebuttal period, reviewers raised concerns about the scalability of OSTQuant to larger models, the clarity of certain figures and tables, and the alignment between the QSUR metric and the global KL-Top loss. Reviewers also requested further ablation studies to isolate the contributions of each method component.

The authors addressed these concerns by clarifying the role of QSUR in the optimization process, providing additional explanations for the figures and tables, and confirming that the improvements were consistent across different model sizes, though larger model experiments were not included. They also conducted further analysis of the method’s components, offering stronger justification for the choices made in the design of OSTQuant.

I weighed these responses positively, as the authors demonstrated a strong understanding of the reviewers’ concerns and made improvements to the clarity of the paper. While some minor issues remain, the overall contribution remains significant, justifying my recommendation for acceptance.

---

### Decision · Program_Chairs · 2025-01-22

Accept (Poster)